# Overcoming $C_{60}$-induced interfacial recombination in inverted perovskite solar cells by electron-transporting carborane

Fangyuan Ye[1,2], Shuo Zhang[1], Jonathan Warby[2], Jiawei Wu[3], Emilio Gutierrez-Partida[2], Felix Lang[2], Sahil Shah[2], Elifnaz Saglamkaya[2], Bowen Sun[2], Fengshuo Zu[4], Safa Shoaee ●[2], Haifeng Wang[3], Burkhard Stiller[2], Dieter Neher ●[2], Wei-Hong Zhu ●[1], Martin Stolterfoht ●[2] ✉ & Yongzhen Wu ●[1] ✉

Inverted perovskite solar cells still suffer from significant non-radiative recombination losses at the perovskite surface and across the perovskite/$C_{60}$ interface, limiting the future development of perovskite-based single- and multi-junction photovoltaics. Therefore, more effective inter- or transport layers are urgently required. To tackle these recombination losses, we introduce ortho-carborane as an interlayer material that has a spherical molecular structure and a three-dimensional aromaticity. Based on a variety of experimental techniques, we show that ortho-carborane decorated with phenylamino groups effectively passivates the perovskite surface and essentially eliminates the non-radiative recombination loss across the perovskite/$C_{60}$ interface with high thermal stability. We further demonstrate the potential of carborane as an electron transport material, facilitating electron extraction while blocking holes from the interface. The resulting inverted perovskite solar cells deliver a power conversion efficiency of over 23% with a low non-radiative voltage loss of 110 mV, and retain >97% of the initial efficiency after 400 h of maximum power point tracking. Overall, the designed carborane based interlayer simultaneously enables passivation, electron-transport and hole-blocking and paves the way toward more efficient and stable perovskite solar cells.

Perovskite-based tandem solar cells are almost exclusively based on inverted (*pin*-type) perovskite cells due to their thin charge transport layers (nm to tens of nm) and absence of high-temperature treatments during fabrication (below 100 °C)[1–8]. Thus, improving the performance of these *pin*-type devices is an important task as this is where perovskites will likely enter the market for renewable energy generation. For record-

level *pin*-type cells it has been shown that the voltage loss is governed by the top electron transport layer (e.g., $C_{60}$ or PCBM), and this is quite irrespective of the involved perovskite composition (Fig. 1a). However, the non-radiative recombination at the perovskite/$C_{60}$ interface and the involved defect states remain fundamentally poorly understood[9–12]. Recent results in this regard have shown that recombination occurs

[1]Key Laboratory for Advanced Materials and Joint International Research Laboratory of Precision Chemistry and Molecular Engineering, Shanghai Key Laboratory of Functional Materials Chemistry, Frontiers Science Center for Materiobiology and Dynamic Chemistry, Institute of Fine Chemicals, School of Chemistry and Molecular Engineering, East China University of Science & Technology, Shanghai 200237, China. [2]Institute of Physics and Astronomy, University of Potsdam, D-14476 Potsdam-Golm, Germany. [3]Centre for Computational Chemistry and Research Institute of Industrial Catalysis, School of Chemistry and Molecular Engineering, East China University of Science & Technology, Shanghai 200237, China. [4]Humboldt-Universitat zu Berlin, Institut fur Physik & IRIS Adlershof, Brook-Taylor Straße 6, 12489 Berlin, Germany. ✉e-mail: martin.stolterfoht@uni-potsdam.de; wu.yongzhen@ecust.edu.cn

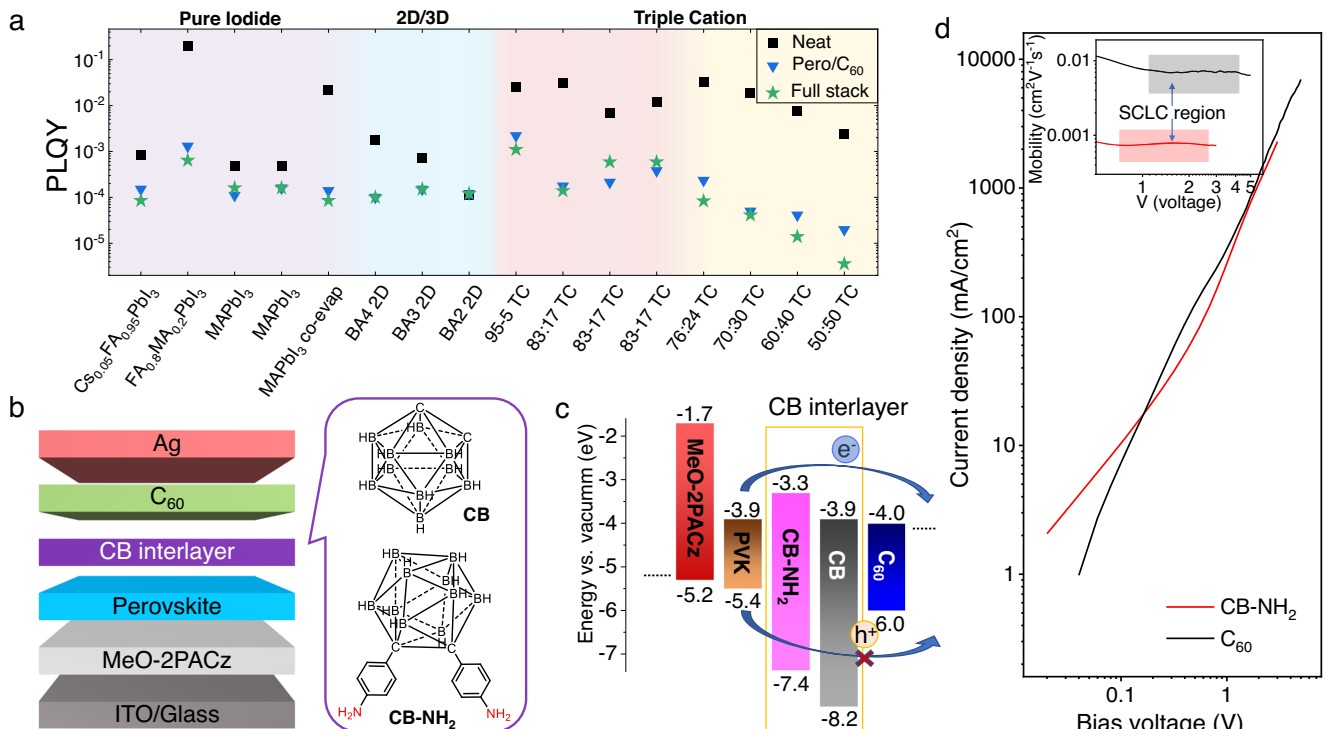

**Fig. 1 | Universal photoluminescence quantum yield (PLQY) losses induced by $C_{60}$, device structure and energy level diagram. a** $C_{60}$-induced recombination losses in various *pin*-type perovskite systems. Data reproduced with permission from Warby, J. et al. Understanding Performance Limiting Interfacial Recombination in pin Perovskite Solar Cells. *Adv. Energy Mater.* 2022, 12, 2103567. 10.1002/aenm.202103567 under a CC BY license https://creativecommons.org/licenses/by/4.0/.[12] $C_{60}$ lowers the PLQY to ≈$1 \times 10^{-4}$–$1 \times 10^{-3}$ of the neat layer, for various perovskite compositions represented by the different shading, such as methyl ammonium lead iodide (MAPbI$_3$), double cation (FA$_{0.8}$MA$_{0.2}$PbI$_3$) and triple-cation perovskites (Cs$_{0.05}$FA$_x$MA$_y$Pb(I$_x$Br$_y$)$_3$) which are denoted as "TC", where the x:y ratio

reflects the molar ration of FAPbI$_3$ versus MAPbBr$_3$, 2D/3D perovskites are denoted by the spacer cation butylammonium (BA) and the number of layers of octahedra between spacer cations, e.g., $n = 4$ is BA$_4$. **b** Device structure and molecule structures of *o*-carborane (CB) as well as phenylamino decorated carborane (CB-NH$_2$). **c** Energy levels based on cyclic voltammetry and literature values. **d** Space-charge limited current measurement of electron-only devices with the configuration of ITO/ZnO/ CB-NH$_2$ (100 nm) or $C_{60}$ (200 nm)/BCP/Ag. The inset shows the mobility variation under different bias voltages. Source data are provided as a Source Data file.

predominantly across the interface[12], likely due to a small energy level offset between the conduction band minimum of the perovskite and the lowest unoccupied molecular orbital (LUMO) level of $C_{60}$[13], or charge transfer states at the interface[12,14]. Thus, it is critical to improve the energy alignment, passivate defects on the perovskite surface or repel minority carriers from the interface. However, to our knowledge, up to now, no study has been able to entirely suppress the interfacial recombination in the presence of the $C_{60}$ layer.

Notwithstanding this point, there have been many attempts to reduce recombination at the perovskite surface and across the perovskite/transport layer interfaces. Noteworthy for *nip*-structured perovskite solar cells (PSCs), Sutuanto et al.[15] showed that the implementation of a two-dimensional (2D) layer (~50 nm thick) on the perovskite surface essentially eliminates the interfacial recombination by preventing minority carriers from reaching the interface between Spiro-OMeTAD and the perovskite. This is due to the nearly lossless interface between the 3D and 2D perovskites and the appreciable thickness of the 2D perovskite layer, which physically separates electrons and holes, thus reducing interface recombination. Indeed, the use of 2D/3D perovskites at the top interface in *nip* cells is very prevalent and enabled high open-circuit voltages ($V_{OCS}$)[16–18]. It, therefore, follows that a similar hole-blocking layer for *pin* cells should lead to reduced voltage losses, however, given the opposite charge polarity, the implementation of these 2D/3D layers in *pin*-type cells has been challenging. Nevertheless, the formation of 2D wide gap perovskites was recently demonstrated in several publications via phenethylammonium (PEA)-based molecules which improved the $V_{OC}$ of *pin*-type perovskite cells[19–21]. A similar surface modification strategy is based on sulfate

(SO$_4$$^{2-}$) and phosphate (PO$_4$$^{3-}$) dissolved in isopropanol or toluene[22,23]. Employed as a surface treatment, these molecules form a thin lead oxysalt layer (PbSO$_4$ and Pb$_3$(PO$_4$)$_2$) on the perovskite surface, which can stabilize under-coordinated bonds on the perovskite surface and improve the stability (T$_{80}$ lifetimes of over 1000 h at 65 °C were reported)[22]. Despite this success, the long-term stability of 2D layers remains a concern, for example, due to the possible interdiffusion and mixing of the 2D/3D perovskite layers as discussed in a previous work[24].

Alternatively, some insulating layers were used to reduce the interfacial recombination without the concern of interdiffusion. For example, commercially available insulating polymers (such as PMMA and PS) have been utilized as interlayers between the perovskite and $C_{60}$ for *pin*-type and for *nip*-type PSCs[25–28]. Moreover, the inorganic insulator LiF has been found effective at reducing the non-radiative recombination loss at the interface for various perovskite systems via the formation of a surface dipole that facilitates the extraction of electrons[9,29]. However, it should be noted that LiF/$C_{60}$ still reduces the photoluminescence quantum yield (PLQY) of the neat layer by a factor of ~10 in the case of triple-cation perovskites[29,30]. Generally, the application of insulating layers leads to the formation of a tunneling contact. However, the application of intrinsically insulating layers limits the ability to block minority charges without compromising the charge transport and thus fill factor (FF) performance. In addition, insulating layers may also cause a loss of the $V_{OC}$ as their low conductivity can lead to a gradient in the quasi-Fermi level of majority charges, resulting in a mismatch between internal and external $V_{OC}$ ("quasi-Fermi level splitting (QFLS)-$V_{OC}$" mismatch)[30–32]. This is, for example, observed

when increasing the thickness of a PMMA interlayer (Supplementary Figs. 1 and 2). Thus, an effective and highly selective electron transport layer requires good hole-blocking properties and good electron transport capacities to simultaneously increase the FF and $V_{OC}$.

In this regard, carboranes are an interesting class of spherical molecules with three-dimensional aromaticity, which are potential electron-transporting materials but have not yet been explored in the field of PSCs. These molecules are composed of carbon, boron, and hydrogen and often form polyhedrons. Polyhedral and spherical molecules with delocalized π-orbitals are ideal for thin film organic transport layers due to their omnidirectional charge transport. They are also tunable as one can functionalize these molecules with various functional groups to tune the electronic structure and other properties of interest[33,34]. Furthermore, they have previously been used in light-emitting diodes[35,36] and as building block of hole-transporting materials in PSCs to improve the charge transfer rate[37].

Herein, we introduced *ortho*-carborane (CB, $C_2B_{10}H_{12}$) as an interfacial layer in low-gap triple-cation *pin*-based PSCs to assist electron extraction and reduce across interface non-radiative recombination. We demonstrate the electron transport potential of functionalized caboranes. We additionally investigated CB functionalized by phenylamino groups (CB-NH$_2$), which are Lewis bases that are reported to passivate perovskite surface defects. Besides, carborane exhibits a large ionization potential thus, it is effective at blocking holes when employed as a selective transport layer. It is also hydrophobic, which effectively improves the water resistance of the perovskite device. Using CB-NH$_2$ as an interlayer we achieve a power conversion efficiency (PCE) of over 23% with low non-radiative voltage losses of 110 mV, in *pin* single junctions, and the PCE retains over 97% of its initial value after 400 h operation under maximum power point (MPP) conditions at 1-sun illumination and room temperature. A variety of experimental investigations indicate that CB-NH$_2$ can overcome the long-standing C$_{60}$-induced interfacial recombination in inverted PSCs.

## Results

### Molecular synthesis and energy levels of carborane

*O*-carborane is commercially available, however, due to its low molecular weight in combination with weak intermolecular forces it has a low sublimation temperature under reduced pressure. This limits its use as an ETL because the thin films can evaporate under high vacuum during the completion of the device (e.g., evaporation of metal electrodes). We therefore chose to synthesize a derivative of *o*-carborane to increase its molecular weight and further modify its electronic/passivation properties. We target a phenylamine functionalized *o*-carborane (structure Fig. 1b, CB-NH$_2$ henceforth) to raise the LUMO by increased conjugation, and reduce electronic traps from previously mentioned DOS broadening via the Lewis base lone pairs on the primary amines[38–41]. The direct addition of functional groups on the carbon sites of *o*-carborane is very challenging. We, therefore, started the molecular synthesis (Supplementary Fig. 3) from dodecahydro-arachno-bis(acatonitrile)decaborane which is available commercially. An electrophilic addition reaction with diphenylacetylene was employed to yield phenyl functionalized CB (in short CB-ph). The amino functionalization of the phenylene group was achieved by an initial nitrification and subsequent reduction reaction to yield CB-NH$_2$. The detailed synthesis, a description of the CB-NH$_2$ molecule and additional molecular structure characterizations (such as nuclear magnetic resonance spectra) can be found in the Supplementary Note 1, and Supplementary Figs. 4–7.

To estimate the energy levels of highest occupied and lowest unoccupied molecular orbitals (HOMO/LUMO), we employed cyclic voltammetry (CV) to measure the reduction potential (LUMO energy levels, Supplementary Fig. 8). Then optical energy gaps ($E_G$) were calculated by the intersection of normalized absorption and emission spectra (Supplementary Fig. 9), and with Tauc plots of the absorption spectra (Supplementary Fig. 10)[42,43]. Finally, the HOMO energy levels were obtained by subtracting the LUMO energy level from the $E_G$ (Supplementary Table 1). As shown in Fig. 1c, the CB-NH$_2$ has different energetics from CB. As predicted, the functionalized phenylamine groups in conjugation with the carborane core significantly raise the HOMO/ LUMO levels. Further in Fig. 1c, we compare the measured energetics with those of FAPbI$_3$ and C$_{60}$ where we observe the much deeper HOMO of the carboranes than C$_{60}$. This should improve its hole-blocking ability as predicted. The energy levels of the perovskite and the electron and hole transport layers (details below) were obtained from photoemission yield spectroscopy and referred from literature[30].

### Electron transport properties of CB-NH$_2$

To evaluate the electron-transporting potential of carborane, we measured electron mobility of CB-NH$_2$ via the space-charge limited current (SCLC) method and compared it with C$_{60}$. We used the following configuration for the electron-only device ITO/ZnO/CB-NH$_2$ (100 nm) or C$_{60}$ (200 nm)/BCP/Ag, which precluded the ability to also assess the electron mobility of CB due to the previously mentioned volatility. As shown in Fig. 1d, the CB-NH$_2$ device exhibits textbook-like SCLC behavior with a clear slope variation from 0–3 V. Using the Mott-Gurney equation of $J = 9\varepsilon_r\varepsilon_0\mu(V_{app}-V_{bi})^2/8L^3$, where $V_{app}$ the applied field and $V_{bi}$ the built-in voltage, $\varepsilon_0$ is the vacuum permittivity ($8.85 \times 10^{-12}$ F m$^{-1}$), $\varepsilon_r$ (3.5) is the dielectric constant, and $L$ is the device thickness, the voltage-dependent mobility can be extracted in the SCLC region. As plotted in the inset picture in Fig. 1d, both the CB-NH$_2$ and C$_{60}$ device reveals the SCLC region (>1 V). This allows us to extract a mobility of $7.3 \times 10^{-4}$ cm$^2$V$^{-1}$s$^{-1}$ for CB-NH$_2$, which is one order of magnitude lower than that of C$_{60}$ ($6.7 \times 10^{-3}$ cm$^2$V$^{-1}$s$^{-1}$). When compared to other non-fullerene organic transport materials from the literature, the mobility of CB-NH$_2$ is middling[44,45]. Nevertheless, when compared to commonly applied interlayer materials such as the widely used Lewis bases (PEAI) and the polymers PS, PMMA, which are basically insulating materials, the electron-transporting capability of CB-NH$_2$ is comparatively high.

### Solar cell fabrication and photovoltaic performance

We then applied CB and CB-NH$_2$ as hole-blocking layers in *pin*-structured triple-cation PSCs using Cs$_{0.05}$(MA$_{0.05}$FA$_{0.95}$)$_{0.95}$Pb(I$_{0.95}$Br$_{0.05}$)$_3$ with an optical bandgap of 1.56 eV as shown from the derivative of the external quantum efficiency (EQE) in the Supplementary Fig. 11. The full device structure was ITO/MeO-2PACz/perovskite/carborane interlayer/C$_{60}$/BCP/Ag. The CB-based interlayers were implemented by spin coating a diluted solution (1 mg/mL in chlorobenzene) on top of the perovskite. Further methods are presented in the Methods.

Figure 2a–c show the PV parameter distribution of the control, CB and CB-NH$_2$- treated solar cells (over 14 cells for each type). The best control cell delivered a PCE of 21.5%, with a $V_{OC}$ of 1.128 V, a $J_{SC}$ of 23.52 mA cm$^{-2}$, and an FF of 81.15%. This provides us with a good starting point for further interfacial improvements and pursuing record-level efficiency. When introducing the CB as an interlayer, we find that the $V_{OC}$ remains unchanged, while there is a statistical improvement in fill factor (FF) and short-circuit current ($J_{SC}$). The best CB-based devices exhibit a slightly lower $V_{OC}$ of 1.120 V, with a $J_{SC}$ of 23.77 mA cm$^{-2}$, and an FF of 81.68%, which represents only a small improvement compared to the control device. Any improvement is surprising given the volatile nature of the CB molecule, therefore we speculate there remains a small amount on the surface.

The introduction of CB-NH$_2$ has a far more profound effect on the device parameters. The *JV* curves of the best-performing cells with and without CB-NH$_2$ are shown Fig. 2d. The champion CB-NH$_2$ treated cell exhibit a PCE of 23.05%, with a high voltage of 1.170 V for the given

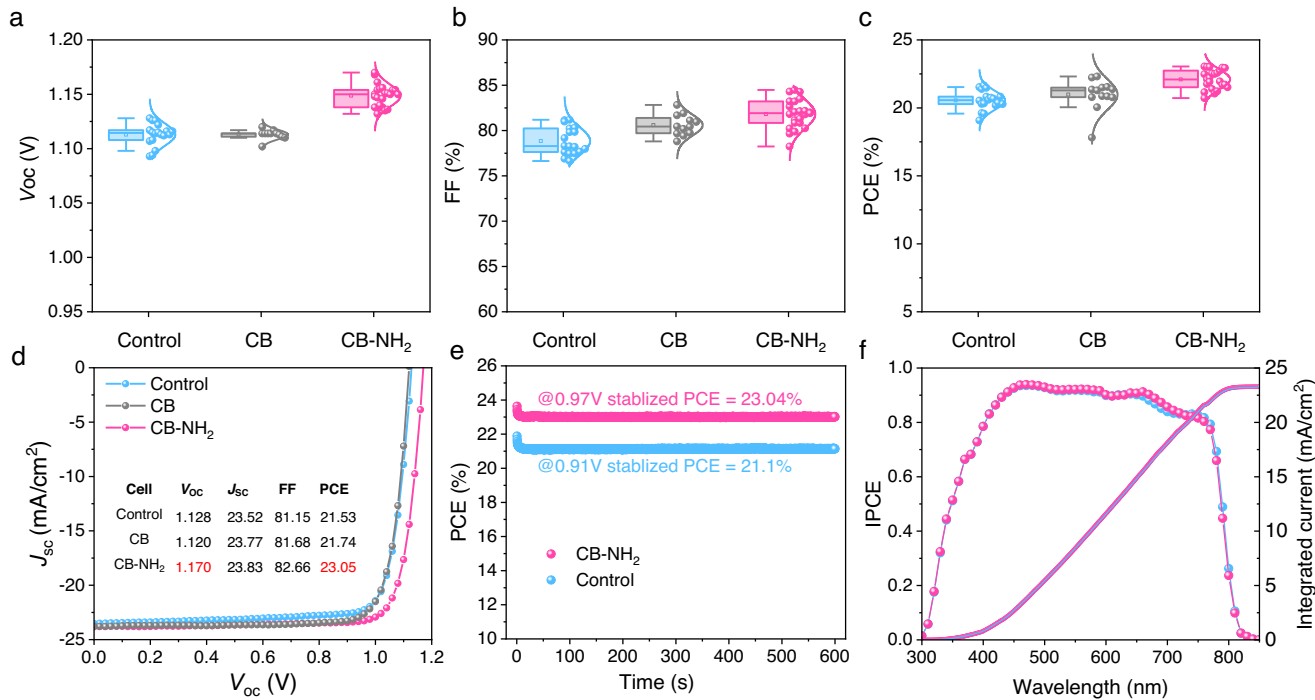

**Fig. 2 | Device performance of CB-NH$_2$, CB treated and control cells.**
**a–c** Parameter distributions and corresponding standard deviations of $V_{OC}$, FF, and PCE for control, CB, and CB-NH$_2$ treated devices, respectively. **d** JV curves of the champion devices. **e** Stable power output results of CB-NH$_2$ based and control cells.

**f** Incident photon-to-current conversion efficiency and the integrated current density results of CB-NH$_2$ treated and control cells. Source data are provided as a Source Data file.

bandgap of the absorber (1.56 eV) and a small hysteresis at a scan speed of 20 mV/s as shown in Supplementary Fig. 12. Based on the EQE spectrum (Supplementary Fig. 13), we calculated the radiative recombination current in the dark ($J_{0,rad}$), which corresponds to a radiative $V_{OC}$ limit of 1.28 eV. Therefore, the non-radiative voltage losses are approximately 110 mV. This is among the smallest $V_{OC}$ deficits for *pin*-structured cells, with only few reports that achieved even smaller voltage losses[23,46,47]. We also measured the steady-state power output (SPO) of the devices at their MPP voltage for 600 s. As shown in Fig. 2e, the stabilized current density of CB-NH$_2$ treated and control devices are 23.73, 23.26 mA/cm$^2$ at a bias of 0.97 and 0.91 V, respectively, confirming a stabilized PCE of 23.04% and 21.10%, which are consistent with JV results. We note that there was a PCE drop in the initial 10 s for both devices, which is likely due to current losses related to ion movement in the perovskite absorber[48,49]. We also note that the integrated current density from the EQE spectrum matches well with the JV results (Fig. 2f).

**Passivation and device operational stabilities**
Highly motivated by the excellent performance improvements rendered by the CB-NH$_2$ interlayer, we proceeded to quantify the device- and passivation stability. We first confirmed that the CB-NH$_2$ molecules do not change the absorption spectrum of the perovskite layer therefore unlikely to alter the perovskite bulk properties (Supplementary Fig. 14). We then performed water contact angle measurements to evaluate the influence of CB-NH$_2$ on the surface properties of perovskite. A larger water contact angle of 75° was determined for the CB-NH$_2$ coated perovskite film, in comparison to the 41° of the bare perovskite (Fig. 3a). The hydrophobicity of the carborane is beneficial for protecting perovskite from the ingress of moisture. Thermal stability is another concern for various interfacial passivation materials. We note that carboranes are regarded as thermally stable molecules that have been used as a building block for high-temperature-resistant materials[50]. A previous study has proven that the passivation effect of

PEAI, one of the most popular passivation materials, deteriorates at elevated temperatures, causing decreased $V_{OC}$ in devices as a result of the formation of the 2D phase[51]. Similarly, LiF has also been suggested to negatively influence device stability[29]. We, therefore, compared CB-NH$_2$ with these two widely used materials at elevated temperatures (50 °C to 100 °C), and measured the PLQY variation to assess the "passivation stability". As shown in Fig. 3b, the CB-NH$_2$ coated perovskite sample shows the highest initial PLQY value, indicating an effective passivation of defects. At higher temperatures, the PLQY values of all samples decreases, e.g., the PEAI sample retains only 27% of its initial value, and this value for LiF sample is 41.2%. For comparison, the CB-NH$_2$ coated film demonstrates a smaller decline, retaining 74% of its initial PLQY. Moreover, we noticed that after cooling down to room temperature, the PLQY of PEAI and LiF samples remained at a similar value as at 100 °C, indicating an irreversible process, while the CB-NH$_2$ sample almost returned to its initial value. We noted that no new diffraction peak was detected in the CB-NH$_2$ sample after annealing, indicating neither the molecule nor any degradation product entered the perovskite and no 2D phase was formed (Supplementary Fig. 15). To quantify the stability, we performed MPP tracking on the devices in a glovebox using a white-light-emitting diode (LED) without UV at 26 °C. The devices comprising CB-NH$_2$ maintained over 97% of the initial PCE (Fig. 3c) under MPP and 1-sun equivalent illumination for 400 h, indicating superb operational stability. In contrast, the control sample retains 89% of its initial efficiency over the same time. The improved operational stability is attributed to effective passivation, which reduces the surface reactivity and retards the degradation rate. We further confirm the improved operational stability during maximum power point tracking under 1-sun equivalent illumination with a white LED in air (30% RH) and at a temperature of 40 °C. As shown in Supplementary Fig. 16, the CB-NH$_2$ devices also demonstrate better stability under these conditions, thus the improvement might be related to the increased hydrophobicity and resistance to moisture.

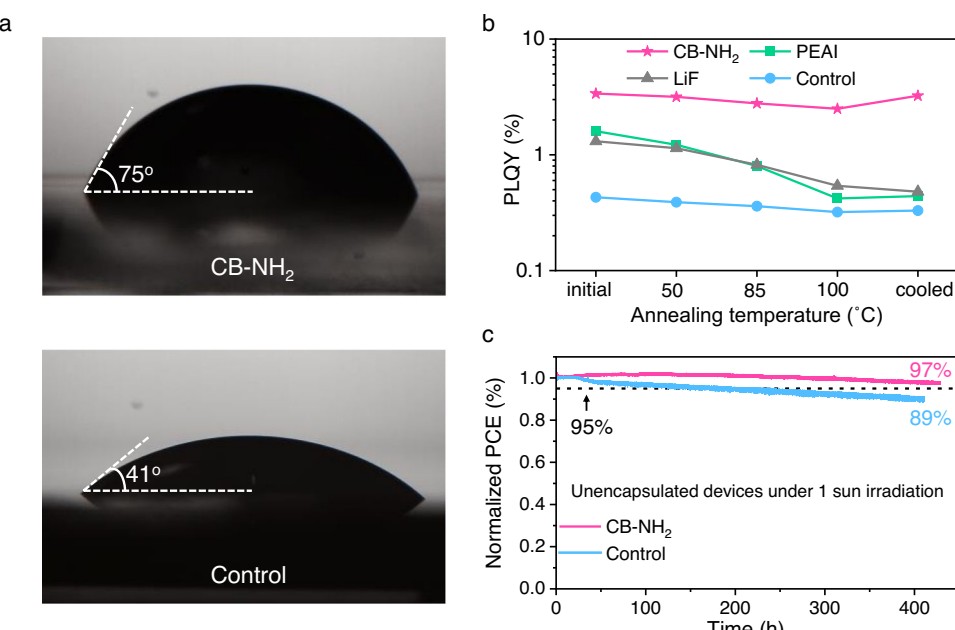

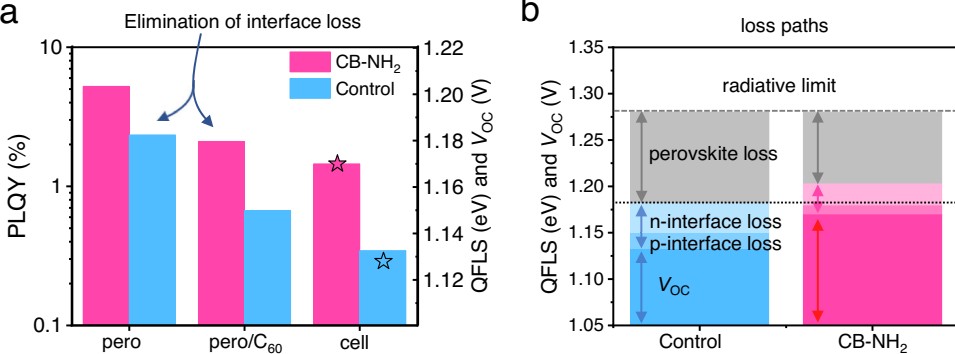

**Fig. 3 | Stability evaluation. a** Contact angle measurement using water for a neat reference and a CB-NH$_2$-treated perovskite. For the CB-NH$_2$ treated samples, the contact angle with water is larger. **b** Passivation stability of CB-NH$_2$, PEAI, LiF on perovskite film under gradually elevated temperature. **c** Maximum power point tracking under a 1-sun equivalent intensity of CB-NH$_2$ based and control PSCs in an inert atmosphere at 26 °C. Source data are provided as a Source Data file.

**Fig. 4 | Photoluminescence quantum yield measurements (PLQY) and internal quasi-Fermi level splitting (QFLS) diagram. a** PLQY and QFLS diagram and V$_{OC}$ from *JV* results for perovskite film, perovskite/C$_{60}$ half stack, and full cell with/without CB-NH$_2$. The almost identical PLQY value between pero and pero/CB-NH$_2$/ C$_{60}$ samples indicates the elimination of interfacial recombination. The bars represent the QFLS results from PLQY measurement and the stars the V$_{OC}$ from *JV* results, respectively. **b** Voltage loss mechanism for the control and CB-NH$_2$-based samples. Source data are provided as a Source Data file.

## Suppression of perovskite/C$_{60}$ interfacial recombination

To understand the simultaneous improvement of the FF and the V$_{OC}$ in case of CB-NH$_2$, we first employed PLQY measurements to analyze the non-radiative recombination in perovskite films, multilayer partial cell stacks, and complete PSCs. Given that it is a non-contact technique, PLQY characterization allows one to decouple the contribution of every layer/interface to the non-radiative recombination loss. We measured different partial cell stacks and compared the results with complete devices to enable insights into the working mechanism of CB-NH$_2$. We use a laser of 520 nm wavelength with a spot size of near 0.5 cm$^2$ to illuminate the samples with a 1-sun equivalent intensity by adjusting the produced current close to the J$_{SC}$ under a standard solar simulator. As shown in Fig. 4a and Supplementary Fig. 17, the PLQY of the bare perovskite on glass is improved from 2.3% to 5.2% due to the CB-NH$_2$ treatment, indicative of surface trap passivation. To correlate the PLQY results with the kinetic properties of photogenerated charges, time-resolved photoluminescence (TRPL) characterization was performed. As shown in Supplementary Fig. 18, a neat perovskite film without charge

transport layers exhibit approximately a mono-exponential decay. In this case, then a Shockley–Read–Hall (SRH) recombination-dominated lifetime ($\tau_{SRH}$) can be deduced to be 1800 and 1000 ns for CB-NH$_2$ treated and control films, respectively. The extended $\tau_{SRH}$ of CB-NH$_2$ coated perovskite films is favorable for carrier collection. When capped with C$_{60}$, the PLQY of the control sample was reduced from 2.3% to 0.67% by a factor of ~4. In contrast, the C$_{60}$ induced PL quenching was substantially reduced for the CB-NH$_2$ treated perovskite/C$_{60}$ stacks (from 5.2% to 2.1%). The PLQY of perovskite/CB-NH$_2$/C$_{60}$ stack is nearly identical to the bare perovskite sample (2.3%), indicating that we overcame the critical C$_{60}$-induced non-radiative recombination upon introducing the CB-NH$_2$ interlayer for this system. To our knowledge, this is the first time that this was achieved for *pin*-type perovskite cells based on a C$_{60}$ ETL. As a result, we obtained a very high PLQY of 1.45% in full device, which is rarely achieved in *pin*-type cells, demonstrating the unique passivation effect of carborane interlayer.

To quantify the open-circuit voltage potential for every individual stack, we calculated the internal electron-hole quasi-Fermi level

**Table 1 | PLQY and QFLS results of perovskite film, perovskite/$C_{60}$ half stack, and full cell with/without CB-NH$_2$**

| Sample | PLQY | QFLS (eV) | $V_{OC}$ (V) |
|---|---|---|---|
| pero/CB-NH$_2$ | 0.0523 | 1.204 | |
| neat pero | 0.0234 | 1.183 | |
| pero/CB-NH$_2$/C$_{60}$ | 0.021 | 1.180 | |
| pero/C$_{60}$ | 0.0067 | 1.150 | |
| cell-CB-NH$_2$ | 0.0145 | 1.170 | 1.170 |
| cell-control | 0.00345 | 1.133 | 1.128 |

splitting (QFLS) from PLQY results by using the equation QFLS = $k_B T^* \ln(\text{PLQY} \times J_G/J_{0,\text{rad}})$, where $J_G$ is the generated current density at 1-sun and $J_{0,\text{rad}}$ is the radiative recombination current in the dark as shown in Supplementary Fig. 13. As shown in Fig. 4a, the neat perovskite exhibits an implied $V_{OC}$ of 1.183 V. This value increases to 1.204 V after the introduction of CB-NH$_2$, which is attributed to the passivation effect on the perovskite surface. The perovskite/C$_{60}$ half stack exhibits a lower QFLS of 1.150 V, while the CB-NH$_2$ treated perovskite/C$_{60}$ sample shows QFLS of 1.180 V, which is nearly equal to the neat perovskite, indicating that the C$_{60}$-induced interfacial recombination is effectively overcome. The PLQY and QFLS values mentioned above are collected in Table 1. Moreover, the difference between the QFLS and $V_{OC}$ of the cells is negligible in both samples (w/ and wo/CB-NH$_2$). This demonstrates spatially flat Fermi levels throughout the device and well-aligned energy levels for majority carriers. Note any misalignment (downhill energy offset for electrons and uphill offset for holes) in combination with a finite interface recombination velocity would readily cause a gradient in the electron/hole quasi-Fermi levels and a QFLS-$V_{OC}$ mismatch. Figure 4b depicts the voltage loss mechanism in the control and CB-NH$_2$-based devices. Under the limitation of thermodynamic radiative which is 1.28 eV for the given bandgap, the perovskite loss and interface loss dominate the voltage loss in both samples whereas the transport loss is negligible (see the identical QFLS and $V_{OC}$ of the full device). After the introduction of CB-NH$_2$, the simultaneously suppressed perovskite loss and interface loss result in the improved $V_{OC}$.

In order to gain more insight into the influence of charge recombination dynamics in complete devices, we also performed transient photovoltage (TPV) measurements under a white LED with an equivalent intensity of 1-sun. As shown in Supplementary Fig. 19, the CB-NH$_2$ treated device exhibits a long lifetime (4.43 μs) which is nearly twice as much as the blank device (2.26 μs). The improved carrier lifetime is consistent with the PLQY results. We note that this value is larger than the bulk SRH lifetime obtained from TRPL, which is likely related to capacitive effects impacting TPV[52,53]. In addition, electrochemical impedance spectroscopy (EIS) measurements were performed to characterize the charge carrier recombination near open-circuit conditions (≈1 V), the series resistance ($R_s$), and recombination resistance ($R_{rec}$) of the CB-NH$_2$ and the reference device. All of these parameters can be obtained from the Nyquist plots intuitively. The equivalent circuits are presented in Supplementary Fig. 20. The semicircle of the CB-NH$_2$-based devices is larger than that of the blank device, indicating a significant reduction in charge carrier recombination rate, attributed to the lower $R_s$ and higher $R_{rec}$. We also estimated the apparent trap density ($n_{trap}$) of the perovskite films by measuring the dark $JV$ profiles of the electron-only devices. As shown and discussed in Supplementary Fig. 21, the current in the ohmic region is smaller for the CB-NH$_2$-based device, which could be correlated with a lower overall trap density and the improved $V_{OC}$[54]. However, the measurable trap density from this experiment is limited by the electrode charge per unit cell volume, which is typically very similar to the apparent trap density in thin film (~500 nm) devices as recently shown in refs. 55, 56.

We then aimed towards an understanding of the impact of our carborane passivation on the ideality factor ($n_{ID}$), fill factor losses, and efficiency potential and therefore investigated the intensity (or voltage) dependence of the non-radiative recombination losses. To this end, we employed three techniques: intensity-dependent quasi-Fermi level splitting, intensity-dependent $V_{OC}$, and injection-dependent electroluminescence (EL). We first measured the intensity-dependent PLQY (iPLQY) of the different stack of samples. Upon converting the PLQY at different laser intensities to the QLFS, we obtain the QFLS as a function of light intensity (Fig. 5a). Furthermore, we construct a pseudo-dark $JV$ curve by plotting the recombination current minus the voltage-independent generation current under AM1.5 G ($J_{SC}$) on the y-axis and the QFLS as $x$ axis. As a result, we create a pseudo-light $JV$ curves which allow us to quantify the implied $JV$ performance of the film including the FF potential in the absence of $R_s$. As shown in Fig. 5a, CB-NH$_2$-based samples demonstrates a holistically higher QFLS value in the complete device, half stack, and bare perovskite. The pseudo-$JV$ curves are shown in Fig. 5b with photovoltaic parameters collected in Table 2. With regard to the ideality factors, in Fig. 5a for the control cell we obtain an $n_{ID}$ of 1.55 while it is 1.43 for the CB-NH$_2$ treated cell. We note that the difference between the pFF on the cells from the pseudo $JV$s reflects the different FF of the devices. Thus, the improved FF of the CB-NH$_2$-based device is largely due to a lower ideality factor rather than improved charge transport. We also measured the iPLQY and corresponding QFLS on perovskite and perovskite/C$_{60}$ stack with and without C$_{60}$ (Fig. 5c, d). The pseudo $JV$s are plotted in Supplementary Fig. 22 and the parameters are shown in Table 2. The implied FF lost from the neat perovskite to perovskite/C$_{60}$ half stack is 1.5% in the control films, while this decline of FF is reduced to 0.4% for the CB-NH$_2$ coated samples. This demonstrates the introduction of CB-NH$_2$ compensates for the compromised FF caused by the deposition of C$_{60}$ due to dark recombination. For the perovskite and perovskite/C$_{60}$ half stacks, we noticed an increase in $n_{ID}$ by 0.15 from the neat perovskite (1.33) to the perovskite/C$_{60}$ film (1.48), whereas for the CB-NH$_2$ treated sample displays an $n_{ID}$ of 1.43 (Fig. 5c, d). This is again consistent with reduced non-radiative recombination between the C$_{60}$ and the perovskite.

We further performed intensity-dependent $V_{OC}$ measurement and conducted injection-dependent EL measurements to further analyze the $n_{ID}$ and pFF as shown in Supplementary Figs. 23 and 24. Both measurements, further confirm the higher pseudo FF, demonstrating the improved FF due to suppressed non-radiative recombination.

## Electron extraction and hole blocking

To reveal the working mechanism of the CB-NH$_2$ layer, we quantified the charge selectivity at the perovskite/C$_{60}$ interface. To this end, we carried out conductive atomic force microscopy (C-AFM) on ITO/MeO-2PACz/Pero/(CB-NH$_2$)/C$_{60}$ films with and without CB-NH$_2$ interlayer. This method has been used by Xu and Caprioglio et al. previsouly[57,58]. In addition, the samples were illuminated with a 5 mW/cm$^2$ white LED to generate electron-hole pairs. To demonstrate the different transport behavior between electrons and holes, we applied a step-by step voltage variation from positive to negative bias to the tip to imitate the process of electron and hole extraction. Electron extraction is imitated when we apply a positive voltage to the tip while hole extraction is imitated when we apply a negative voltage. Figure 6 demonstrates that the CB-NH$_2$ treated samples exhibits a higher current in the positive bias region, whereas the current is lower at a negative bias with respect to the control sample. To be specific, when 1 V bias is applied to the tip, CB-NH$_2$ treated samples approach an average current of −0.72 nA, while the control sample exhibits a lower average current (−0.2 nA). In contrast, CB-NH$_2$ treated samples exhibit a lower current when −1 V is applied to the tip (0.02 nA) compared to the control sample (0.07 nA). The averaged values were obtained from area-averaged profiles. We note that the isolated spikes could be related to the carrier diffusion

 

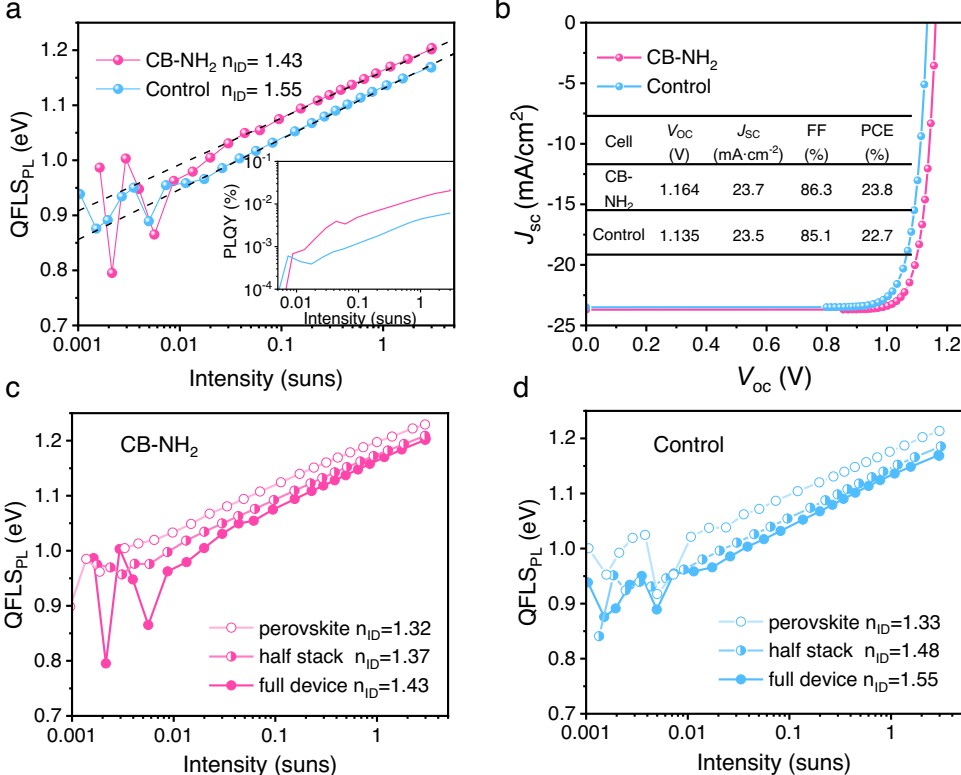

**Fig. 5 | Pseudo-*JV* and efficiency potential. a** Intensity-dependent quasi-Fermi level splitting (QFLS) measurement. **b** Pseudo *JV* curves from intensity-dependent QFLS of the full devices. Intensity-dependent QFLS measurements for neat perovskite, perovskite/$C_{60}$ half stack, and full devices of **c** CB-NH$_2$-based sample and **d** control sample, respectively. The $n_{ID}$ is smaller for samples with carborane, indicating less non-radiative recombination. Source data are provided as a Source Data file.

pathway or the morphology. Moreover, all samples have relatively lower current value in the negative region than in the positive region, which is consistent with the electron-selective transport through the $C_{60}$ layer. Overall, C-AFM clearly demonstrates that the CB-NH$_2$ layer increases electron extraction while decreasing hole extraction. The improved electron extraction is also consistent with the improved FF and decreased trap density observed with PL. Note, due to the low light intensity ($5\,mW/cm^2$) the measurement is in essence a dark *JV* measurement, and the low values of the current are limited by the small radius of the tip of 35 nm. Therefore, the current signal measured from C-AFM is not comparable to the $J_{SC}$ from *JV* curves.

### Contribution from carborane and amino moiety to device performance

In order to investigate the contribution from the carborane and amino group moiety in CB-NH$_2$ to the improvement of device performance, we also studied the intermediate phenyl functionalized carborane (CB-ph) without the amino group because the bare carborane suffers from

sublimation on the perovskite layer. As can been seen in Supplementary Fig. 25, CB-ph leads to a significant FF improvement (~2%) but only slight $V_{OC}$ improvement (~10 mV). This indicates that the carborane alone reduces non-radiative recombination less effectively than CB-NH$_2$ but has a significant effect on the electron extraction which is consistent with the C-AFM result. The slight $V_{OC}$ improvement can be further confirmed by the improved PLQY of the ITO/perovskite/$C_{60}$ stack with the CB-ph interlayer but identical PLQY of the ITO/perovskite with and without the CB-ph interlayer (Supplementary Fig. 26). These results suggest that there is a small contribution (~10 mV) of the carborane moiety to the total $V_{OC}$ gain of the CB-NH$_2$ functionalized device (~50 mV) which we attribute to the improved hole-blocking ability of the carborane. However, the effect of the amino group is more significant. As for the FF improvement, our results indicate a more significant contribution of the carborane moiety (~2%) to the total gain (~4%), which suggests that the improvement originates partially from the hole-blocking and electron extraction ability of the CB molecule. Nevertheless, considering that the packing and adsorption ability of the CB-ph and the CB-NH$_2$ molecules are likely different, the exact contribution of the carborane moiety is difficult to quantify. Given that the performance of the CB-NH$_2$ device is optimum, we can conclude that the carborane and the amino group are working synergistically and are indispensable.

### Surface passivation and Pb-CB-NH$_2$ interaction

To further explore the passivation mechanism of CB-NH$_2$, we examined the intermolecular interactions between CB-NH$_2$ and perovskite. We conducted Fourier transform infrared spectroscopy (FTIR) and X-ray photoelectron spectroscopy (XPS) measurements to study the interactions between CB-NH$_2$ and perovskite moieties. We dissolved CB-NH$_2$ and PbI$_2$ in DMF solvent and then dried the solution to obtain a

**Table 2 | Parameters of pseudo *JV* curves constructed based on iPLQY from perovskite film, perovskite/$C_{60}$ half stack, and full cell with/without CB-NH$_2$**

|  | cell | $V_{OC}$ (V) | $J_{SC}$ (mA cm$^{-2}$) | FF (%) | PCE (%) |
|---|---|---|---|---|---|
| CB-NH$_2$ | Device | 1.164 | 23.7 | 86.3 | 23.8 |
|  | Half stack | 1.177 | 23.7 | 86.8 | 24.1 |
|  | Perovskite | 1.202 | 23.7 | 87.2 | 24.8 |
| control | Device | 1.135 | 23.5 | 85.1 | 22.7 |
|  | Half stack | 1.146 | 23.5 | 85.6 | 23.0 |
|  | Perovskite | 1.181 | 23.5 | 87.1 | 24.1 |

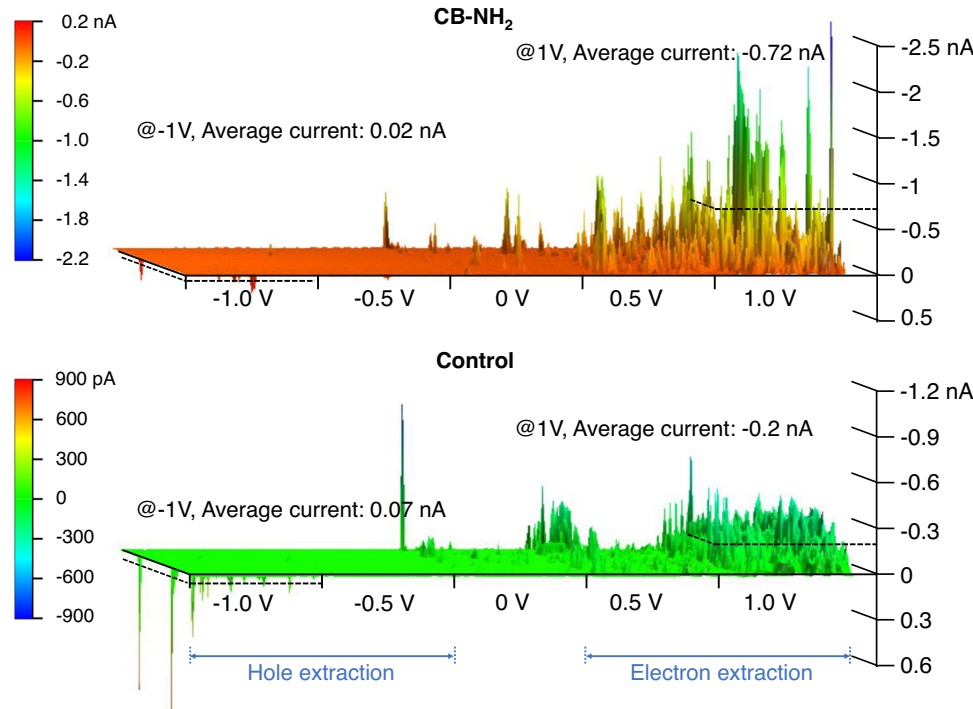

**Fig. 6 | Improved electron extraction and hole blocking at perovskite/$C_{60}$ interface.** 3D conductive-AFM images (5 × 5 µm) of the samples with a structure of ITO/MeO-2PACz/Pero/CB-NH$_2$/ $C_{60}$ (CB-NH$_2$ treated) and ITO/MeO-2PACz/Pero/ $C_{60}$ (control). The images represent a step-by-step applied voltage variation from −1.0 V applied to the tip (left) for hole extraction to 1.0 V applied to the tip (right) for electron extraction.

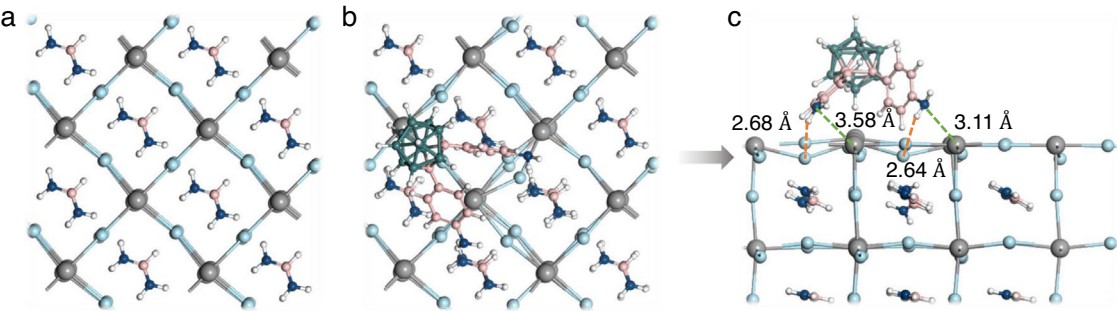

**Fig. 7 | Interaction between CB-NH$_2$ and perovskite from DFT calculations.** Top views of PbI$_2$-terminated perovskite surface **a** without and **b** with CB-NH$_2$. **c** Side views of the perovskite surface adsorbed with CB-NH$_2$. Notes: the length of N-H···I/ N-Pb are depicted. Color scheme: Pb: gray; I: light blue; C: pink; N: dark blue; B: green; H: white.

mixed powder[40,59]. The FTIR spectra of the mixture and pure CB-NH$_2$ powders, displayed in Supplementary Fig. 27, reveal that the addition of PbI$_2$ results in the variation of the $v$(N-H) interaction. Given that the lone pairs of N contained functional groups, especially in the amino group, can interact with Pb$^{2+}$, we speculate the shift of the $v$(N-H) peak originates from the interaction between the N atom and Pb$^{2+}$. We further performed XPS measurements on perovskite films with/without CB-NH$_2$ to study the interaction between the interlayer and the perovskite. As shown in Supplementary Fig. 28, the Pb signals between 136 eV and 146 eV binding energy demonstrate 2 main peaks, which can be ascribed to the excited electrons of the $4f_{7/2}$ and $4f_{5/2}$ orbitals. Apparently, the signal of the perovskite film with CB-NH$_2$ shows a visible shift towards lower binding energies, indicating the interaction between under-coordinated Pb and CB-NH$_2$. Since the under-coordinated Pb$^{2+}$ on the perovskite surface are typically active sites for interfacial reactions that act as electronic traps[60,61], the interaction between CB-NH$_2$ and Pb might explain the decreased surface trap density, as well as the improved operational stability. We note that the

detected B 1s signal proves the existence of CB-NH$_2$. The observed interaction is likely related to the improvement of PL intensity and lifetime for neat perovskite.

To obtain more atomistic information regarding to the interaction between the perovskite and CB-NH$_2$ density functional theory (DFT) simulations were performed. We choose α-FAPbI$_3$ perovskite and a (001) PbI$_2$-terminated surface as model system in accordance with the studied perovskite. A detailed description of the DFT calculations can be found in Supplementary Note 2. Figure 7a, b displays a top view of the perovskite and the perovskite/CB-NH$_2$ at the lowest energy. Figure 7c, shows a cross-sectional view highlighting that distinct hydrogen bonds are formed between N and H from the CB-NH$_2$ molecule and I from the perovskite surface with a bond length of 2.68 and 2.64 Å, respectively. Such hydrogen bonds with iodide ions have been shown to reduce the trap states and suppress iodide migration, thus improving device performance and long-term stability[62–64]. Moreover, the simulations reveal a very short bond length of Pb-N in both phenylamino groups at 3.58 and 3.11 Å, respectively, indicating CB-NH$_2$

passivates the under-coordinated Pb atoms on the perovskite surface. These short bond lengths result in an extraordinarily high adsorption energy of $-3.42$ eV[64-66]. To further corroborate these findings, we calculated the iodine Frenkel (interstitial/vacancy pair) defect formation energy (DFE) and we find an increased DFE by ~0.2 eV upon the CB-$NH_2$ incorporation (Supplementary Fig. 29), indicating reduced creation and diffusion of iodide defects. Based on the above results, we can draw the conclusion that the synergistic bonding effect from N-H···I and N-Pb realize a strong interaction between the perovskite and CB-$NH_2$, which is beneficial for simultaneously passivating surface defects and improving the surface stability.

## Discussion

In this work, we demonstrate effective control of non-radiative recombination losses at the perovskite/$C_{60}$ interface by implementing a carborane-based interlayer in *pin*-type PSCs. This is achieved upon the introduction of a phenylamine functionalized carborane derivative CB-$NH_2$, which enables efficient defect passivation and selective electron transport with mobility of $7 \times 10^{-4}$ cm$^2$/Vs. The inverted PSCs with CB-$NH_2$ incorporated as an interfacial layer reach high PLQYs of 1.4% and PCEs over 23%, with non-radiative voltage losses as low as 110 mV. By analyzing the PLQY and the QFLS of different partial cell stacks, we demonstrate that the implementation of CB-$NH_2$ as an interlayer nearly eliminates the $C_{60}$-induced interfacial recombination, which has not yet been achieved with other interfacial modifications. Based on a series of experimental methodologies, we demonstrate that the concomitantly improved $V_{OC}$ and FF with CB-$NH_2$ is a result of the passivation of the perovskite surface, reduced non-radiative recombination across the perovskite/$C_{60}$ interface combined with an increased charge carrier selectivity. The CB-$NH_2$ interlayer also improves the thermal and operational stability of the complete devices, which is attributed to the stabilization of defects on the perovskite surface and the hydrophobic nature of CB-$NH_2$. This work addresses one of the most critical issues of inverted PSCs and demonstrates the potential of carborane derivatives as interfacial and electron transport materials in perovskite-based opto-electronic devices. As such, we believe that future molecular design and optimization will pave the way for a particular class electron transport materials based on carboranes.

## Methods

### Materials and reagents
The raw material dodecahydro-arachno-bis(acatonitrile)decaborane was purchased from Zhengzhou Yuanli Biological Technology Co., Ltd, China, Diphenylacetylene was purchased from Shanghai Macklin Biochemical Co., Ltd, China. $PbI_2$, $PbBr_2$, MABr, MACl, PEAI, BCP, and MeO-2PACz were purchased from TCI. Patterned indium tin oxide (ITO) glass, FAI, C60, and PCBM were purchased from Advanced Election Technology Co., Ltd. CsI was purchased from Sigma-Aldrich. DMSO, DMF, ethanol, chlorobenzene, and isopropanol were purchased from Acros.

### PSCs fabrication
Patterned ITO-coated glass substrates were cleaned sequentially for 15 min in suds, deionized water, ethanol, and acetone in ultrasonic bath. Then the substrates were treated with ultraviolet ozone (UV-Ozone) for 30 min and transferred to a nitrogen-filled glove box. All subsequent procedures were done in nitrogen-filled glove boxes. The solutions of MeO-2PACz (1 mmol mL$^{-1}$ in ethanol) were spin-coated on the ITO substrates at 3000 rpm for 30 s, followed by annealing at 100 °C for 10 min. The FA-based triple-cation perovskite precursor solution with a formula of $Cs_{0.05}(MA_{0.05}FA_{0.95})_{0.95}Pb(I_{0.95}Br_{0.05})_3$ was prepared by mixing two 1.5 M $FAPbI_3$ and $MAPbBr_3$ perovskite solutions in DMF:DMSO (4:1 volume ratio, v-v), as well as the solution of CsI (1.5 M) and MACl in DMSO (20 mol% MACl were added into the

precursor solution for better crystallization). The triple-cation perovskite films were deposited by spin-coating at 4000 rpm for 40 s (5 s for acceleration). 5 s before the end of the procedure chlorobenzene (300 μL) was dispensed onto the still spinning substrate in one continuous flow which lasts for approximately 1 second, following by annealing at 100 °C for 1 h. Subsequently, the carborane molecules were spin-coated from 1 mg/mL solution in chlorobenzene at an rpm of 6000 (3 s for acceleration). Then, the samples were transferred to an evaporation chamber where $C_{60}$ (Sigma-Aldrich, 30 nm) at 0.2 Å/s, 8 nm BCP (Sigma-Aldrich) at 0.2 Å/s and 100 nm copper (Sigma-Aldrich) at 0.6 Å/s were deposited under vacuum ($p = 10^{-7}$ mbar).

### Characterization and measurements
**Absolute photoluminescence measurements.** Excitation for the PL measurements was performed with a 520 nm CW laser (Insaneware) through an optical fiber into an integrating sphere. The intensity of the laser was adjusted to a 1-sun equivalent intensity by illuminating a 1 cm$^2$-size PSC under short-circuit and matching the current density to the $J_{SC}$ under the sun simulator (22.0 mA/cm$^2$ at 100 mW cm$^{-2}$, or $1.375 \times 10^{21}$ photons m$^{-2}$ s$^{-1}$). A second optical fiber was used from the output of the integrating sphere to an Andor SR393iB spectrometer equipped with a silicon CCD camera (DU420A-BR-DD, iDus). The system was calibrated by using a calibrated halogen lamp with specified spectral irradiance, which was shone into to integrating sphere. A spectral correction factor was established to match the spectral output of the detector to the calibrated spectral irradiance of the lamp. The spectral photon density was obtained from the corrected detector signal (spectral irradiance) by division through the photon energy ($h\nu$), and the photon numbers of the excitation and emission obtained from numerical integration using MATLAB.

**Electroluminescence measurements (EL).** The absolute EL intensity was measured with a calibrated Si photodetector and a Keithley 485 pico Ampere meter. The detector (with an active area of ~2 cm$^2$) was placed directly in front of the device (0.13 cm$^2$), and the total photon flux was evaluated considering the emission spectrum of perovskite and silicon sub-cell, their relative intensities from the previously measured relative EL spectra and the external quantum efficiency of the detector. A slight underestimation of the EQE$_{EL}$ ($\approx 1.08 \times$), originating from an additional glass encapsulation, and some photons that escaped to the side and were not detected was compensated. A forward bias was applied to the cell using a Keithley 2400 source meter, and the injected current was monitored. Measurements were conducted with a home-written LabVIEW routine. Typically, the voltage was increased in steps of 20 mV, and the current stabilized for typical 1 s at each step.

**Intensity-dependent $V_{OC}$ measurements.** Steady-state intensity-dependent $V_{OC}$ (i-$V_{OC}$) was measured under mimicked AM1.5 G illumination (Oriel class AAA Xenon lamp-based sun simulator), appropriated neutral density filters, and a Keithley2400. The samples were irradiated by using a 445 nm continuous wave laser (Insaneware). A continuously variable neutral density filter wheel (ThorLabs) was used to attenuate the laser power (up to OD 6). The light intensity was thereby simultaneously measured with a silicon photodetector and a Keithley 485 to improve the accuracy of the measurement. For each intensity, the $V_{OC}$ is measured for ~2 s before the $J_{SC}$ is measured for the same time. The filter wheel is then moved to the next position and the routine is repeated. A homemade lab view program controlled the variable neutral density filter wheel and measured the $V_{OC}$ using a Keithley2400.

**Current density-voltage characteristics.** *JV*-curves were obtained in a 2-wire source-sense configuration with a Keithley 2400. An Oriel

class AAA Xenon lamp-based sun simulator was used for illumination providing approximately 100 mW cm$^{-2}$ of simulated AM1.5 G irradiation and the intensity was monitored simultaneously with a Si photodiode. The exact illumination intensity was used for efficiency calculations, and the simulator was calibrated with a KG5-filtered silicon solar cell (certified by Fraunhofer ISE). The temperature of the cell was fixed to 25 °C and a voltage ramp of 20 mV/s was used. The active area of tested cells was 12 mm$^2$ as defined from the area of overlap of the electrodes. To measure the *JV*-characteristics an illumination mask with an area of 9 mm$^2$ was used. The spectral mismatch factor ($S_M$) was calculated to be 0.982 for the optimized devices with EQE spectra presented in Fig. 2f, however, the obtained current has not been upscaled by 1/$S_M$.

**Conductive atomic force.** Conductive Atomic Force Microscopy was performed with a Solver NT-MDT instrument with a hardware linearized 100 mm scanner and scanning tip. The measurements were performed in contact mode by measuring both spreading resistance and topography. The tip used was a platinum NSG10/Pt.

**Other Measurements.** The $^1$H NMR spectra were recorded with the Bruker AM 400 spectrometer. The cyclic voltammograms were measured by using a CHI660E electrochemical workstation (Chenhua Co. Ltd, Shanghai, China) in a three-electrode cell. X-ray diffraction measurements were obtained by a Rigaku Ultima IV (Cu Ka radiation, $\lambda = 1.5406$ Å) in the range of 5°–90° (2θ). The time-resolved photoluminescence decay lifetimes were determined with the single photon counting technique by Edinburgh FLS1000 spectrometer. Transient photovoltage test was conducted under a white-light (LED) which was adjusted to produce the $V_{OC}$ at 1-sun conditions (1-sun equivalent), and the laser pulses (532 nm, 2 Hz) to measure the decay of transient photovoltage signals. EQE measurements were measured by Newport-74125 system (Newport Instruments). The operational stability of the unencapsulated PSCs was performed at the MPP tracking under continuous white-light LED array illumination in N$_2$ atmosphere, the light intensity was calibrated to achieve the same $J_{SC}$ from the PSCs measured under a standard solar simulator (AM1.5 G, 100 mW cm$^{-2}$).

**Reporting summary**
Further information on research design is available in the Nature Research Reporting Summary linked to this article.

## Data availability
The data generated in this study are provided in the Supplementary Information/Source Data file. Source data are provided with this paper.

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

## Acknowledgements

This work was supported by the National Natural Science Foundation of China (22179037), Shanghai Municipal Science and Technology Major Project (2018SHZDZX03, 21JC1401700), Programmer of Introducing Talents of Discipline to Universities (B16017), and Fundamental Research Funds for the Central Universities. We acknowledge HyPerCells (a joint graduate school of the University of Potsdam and the Helmholtz-Zentrum Berlin) and the Deutsche Forschungsgemeinschaft (DFG, German Research Foundation)—SPP 2196 (project number 423749265-SURPRISE and 424709669-HIPSTER)—SFB951 (project number 182087777) for funding. We also acknowledge financial support from the Federal Ministry for Economic Affairs and Energy within the framework of the 7th Energy Research Program (P3T-HOPE, 03EE1017C). Additional funding came from the Deutsche Forschungsgemeinschaft (DFG, German Research Foundation) – Projektnummer 491466077. M.S. further acknowledges the Heisenberg program from the Deutsche Forschungsgemeinschaft (DFG, German Research Foundation) for funding—project number 498155101. Y.W. thank the Research Center of Analysis and Test of East China University of Science and Technology for the help on NMR and FTIR analysis.

## Author contributions

F.Y. conceived and designed the experiments, synthesized the molecules, analyzed and interpreted data, conceptualized the study, and drafted the manuscript; S.Z. contributed to the perovskite optimization and performed device characterization. Jo.W. helped to characterize and analyze PMMA-containing devices. Ji.W. and H.W. performed DFT simulations and corresponding analysis. Jo.W., E.G-P., and F.L. contributed to PL, pseudo-$JV$ characterization, EL, and $V_{OC}$-suns experiments and data analysis. Sahil.S. contributed to device fabrication and characterization, E.S., Bo.S., and Safa.S. contributed to SCLC measurements and corresponding data analysis. F.Z. contributed to the analysis of energy level measurements and XPS. Bu.S. performed C-AFM measurements and data analysis. D.N. and W.Z. contributed to the molecule development, interpretation of all experimental results, and organic molecule characterization. M.S. and Y.W. contributed to data analysis, and interpretation of the results, and conceptualized the study. All authors contributed to proofreading and revising the manuscript.

## Funding

## Competing interests

The authors declare no competing interest.
