## [Peer Review File · Nature Communications]

Overcoming C60-induced interfacial recombination in inverted perovskite solar cells by electron transporting carboraneREVIEWER COMMENTS

Reviewer #1 (Remarks to the Author):

This manuscript reports a novel interface passivation layer for inverted perovskite solar cells with C60 electron transport layers. C60 is by far the most widely used ETL material in inverted (pin) perovskite cells, but it is also responsible for the relatively poor voltage performance of these cells relative to nip cells. This work clearly demonstrates that non-radiative recombination at the C60/perovskite interface is the cause of this voltage loss, and demonstrates an effective solution with the functionalized carborane passivation. The passivation mechanism is carefully investigated using both experimental and theoretical (DFT) methods. The improved stability of the passivated cells with the hydrophobic interlayer is also promising.

The manuscript is well written, the experimental and theoretical analysis is thorough and clearly presented, and the conclusions are well supported by the evidence. This work is likely to be of significant interest to the perovskite research community and therefore I recommend it for publication with only minor revisions as listed below:

1. The introduction (line 85) cites Peng [ref 21] as demonstrating PMMA for passivating the perovskite/C60 interface. This is not quite correct: ref 85 only reported results for pin structured cells passivated by PMMA.
2. Line 156: "The optical energy gaps (EG) were calculated by the intersection of normalized absorption and emission spectra". This method for estimating the optical bandgap is not commonly used in perovskite literature, although it may be standard in other fields. I suggest that the authors either provide a suitable reference, or else provide a brief explanation/justification in the SI.
3. Was the cell temperature controlled during the MPP stability tracking measurements in Fig 3c? If so, the temperature should be specified. Also related to Fig 3c, were multiple cells tested for stability, or only one of each type? Given the typical performance spread of individual perovskite cells, stability conclusions based on single device measurements are not very reliable. The authors should provide stability data for multiple cells or if not, they should comment on the confidence of their conclusions.
4. Line 333: trap densities extracted from SCLC measurements are quoted to three significant figures. Is this level of precision appropriate given sample-sample variation and the assumptions required to extract these values?

5. Line 391: The conductive AFM results show >3x increase in photocurrent for CB-NH₂ samples at 1V bias. It is not obvious whether there is a quantitative relationship between these results and the device performance. In particular, why is the difference in (averaged) nanoscale photocurrent so much larger than the photocurrent observed at the device level?

6. There are multiple typos and a few grammatical errors throughout the manuscript. Careful proof-reading is recommended before final submission.

Reviewer #2 (Remarks to the Author):

For the "Overcoming C60-induced interfacial recombination in inverted perovskite solar cells by stable electron transporting carborane", the authors introduced carborane as electron transport layer in perovskite solar cell structure. However, the manuscript lacks novelty on the use of carborane and progressive discussion of defect passivation in this study and needs further investigation and improvement of the experimental data. Therefore, this manuscript is insufficient to be published in Nature Communications.

1. Although carborane is a material that exhibits unique properties, there are more cases of perovskite solar cells applied in previous studies than the authors mentioned. Therefore, authors should add references related to perovskite photovoltaics applied with carborane.

2. It is hard to understand that the carborane did not affect the perovskite even though the carborane was deposited on top of the perovskite. Since the carborane has a great ability to absorb neutrons, the carborane can control charge flows and ion movements. As the author mentioned, if only a small amount of carborane remains on top of the perovskite, is the carborane very poorly coated on the perovskite? If coating properties are not secured, it would be a meaningful result to use a material other than CB-NH₂ with an ammonium functional group. Therefore, the authors should present the special properties of carborane alone.

3. Moreover, it has been reported in previous studies that inhibition of non-radiative recombination has a great effect on J_{sc} as well as V_{oc}. In this study, CB-NH₂ reduced the loss occurring at the interface through defect passivation. However, defect states will also impede the flow of photogenerated charges. Why is the effect at J_{sc} smaller than the effect at V_{oc}?

Reviewer #3 (Remarks to the Author):

The authors report a novel interlayer material, which helps to achieve efficient and stable inverted perovskite solar cells. In detail, o-carborane decorated with phenylamino groups (CB-NH₂) was introduced on the perovskite surface to eliminate the non-radiative recombination loss across the perovskite/C60 interface. The results are interesting although some of the claims along with the work novelty are overrated. This manuscript shows almost all the data I want to see. Therefore, I suggest accepting the manuscript after addressing the following concerns.

1. The authors introduced a novel material CB-NH₂, so is it the unique advantage of CB or the amino group that plays the leading role? The authors claim that CB-NH₂ can not only passivate surface defects (due to bonding effect from N-H...I and N-Pb), but also block holes (due to the type-I band alignment). That is to say, CB-NH₂ can reduce both the non-radiative recombination assisted by surface defects and the carrier quenching at the perovskite/C60 interface. Which effect contributes more to reducing VOC losses? Since the synthesized intermediates contain CB-ph and CB-NO₂ (almost no bonding effect with perovskite), I wonder how they affect the device performance.
2. As type-I band alignment (as shown in Fig. 1c), why does perovskite/CB-NH₂ increase electron extraction?
3. "As shown in Fig. 2e, the stabilized current density of control and CB-NH₂ treated cell are 23.73, 23.26 mA/cm² under a bias of 0.97 and 0.91V, respectively, confirming a stabilized PCE of 23.04% and 21.10%, which are consistent with JV results." The data for control and CB-NH₂ treated cell were reversed.
4. "To quantify the stability, we performed MPP tracking on the devices in a glovebox... The improved operational stability is attributed to... as well as the increased hydrophobicity and resistance to moisture." Stability results obtained in the glovebox should not be related to moisture.
5. The caption of Fig. S15 is incorrect. In addition, the resolution of light intensity up to 0.0001 sun is incredible. Please provide the corresponding J-V curves of the devices under different light intensity in Fig S15.

Reviewer(s)' Comments to Author and the Point-to-Point Response:

Reviewer: 1

Recommendation: Publish after minor revisions (as noted below).

Comments: This manuscript reports a novel interface passivation layer for inverted perovskite solar cells with C60 electron transport layers. C60 is by far the most widely used ETL material in inverted (pin) perovskite cells, but it is also responsible for the relatively poor voltage performance of these cells relative to nip cells. This work clearly demonstrates that non-radiative recombination at the C60/perovskite interface is the cause of this voltage loss, and demonstrates an effective solution with the functionalized carborane passivation. The passivation mechanism is carefully investigated using both experimental and theoretical (DFT) methods. The improved stability of the passivated cells with the hydrophobic interlayer is also promising.

The manuscript is well written, the experimental and theoretical analysis is thorough and clearly presented, and the conclusions are well supported by the evidence. This work is likely to be of significant interest to the perovskite research community and therefore I recommend it for publication with only minor revisions as listed below:

Thank you very much. We are delighted to receive such a positive feedback.

1. The introduction (line 85) cites Peng [ref 21] as demonstrating PMMA for passivating the perovskite/C60 interface. This is not quite correct: ref 85 only reported results for pin structured cells passivated by PMMA.

Thank you for pointing out this improperly cited reference in the manuscript in line 85, we now replaced ref.21 with ref.12¹ Ref. 21 was cited at the following sentence in the introduction: "For example, commercially available insulating polymers (such as PMMA and PS) have been widely utilized as interlayers between the perovskite and C₆₀ for *pin*-type and for *nip*-type PSCs²¹⁻²⁴."

2. Line 156: "The optical energy gaps (EG) were calculated by the intersection of normalized absorption and emission spectra". This method for estimating the optical bandgap is not commonly used in perovskite literature, although it may be standard in other fields. I suggest that the authors either provide a suitable reference, or else provide a brief explanation/justification in the SI.

We agree that this method is not often used for perovskites, however, it can be applied to determine the optical bandgap of organic semiconductors such as CB-NH₂^{6,7}. In addition, we now use Tauc plots of the absorption spectra to estimate the bandgap of CB-NH₂ (although less often employed for organic semiconductors, see ref.⁸). The optical band gap from the Tauc plot is 3.96 eV, which is close

to what we get from the intersection of normalized absorption and emission spectra (4.10 eV). We have revised the following sentence in the manuscript and added the corresponding reference: "Then optical energy gaps (E_G) were calculated by the intersection of normalized absorption and emission spectra (Supplementary Fig. S5), and with Tauc plots from the absorption spectra (Supplementary Fig. S6)^{6,7}"

Supplementary Fig. S6. Tauc plots of the absorption spectra of CB-NH₂, the optical bandgap is estimated to 3.96 eV.

3. Was the cell temperature controlled during the MPP stability tracking measurements in Fig 3c? If so, the temperature should be specified. Also related to Fig 3c, were multiple cells tested for stability, or only one of each type? Given the typical performance spread of individual perovskite cells, stability conclusions based on single device measurements are not very reliable. The authors should provide stability data for multiple cells or if not, they should comment on the confidence of their conclusions

That's a good point. Because of technical limitations, no temperature controller was used during MPP setup. However, we monitored the temperature on the holder with an infrared sensor and we found that it is around 26 °C. We now added this information to the caption of **Figure 3**. To address the question of reproducibility, we repeated the MPP measurement at the lab in Shanghai, which is in air (30% RH) and temperature at 40 °C. The result is shown below; the unencapsulated CB-NH₂ based device maintains 89% of the initial PCE, compared the 79% of the control device after ~350 h of operation which is similar to the result in the manuscript. This result confirms the consistently improved stability upon addition of CB-NH₂ as reported in the manuscript. We now added this result in the manuscript and the following statement: "We further confirm the improved operational stability during maximum power point tracking under 1 sun equivalent illumination with a white LED in air (30% RH) and at a temperature of 40 °C. As shown in **Supplementary Fig. S12**, the CB-NH₂ devices also demonstrate a better stability under these conditions. Therefore, the improvement might be also related to the increased hydrophobicity and resistance to moisture."

Supplementary Fig. S12. Maximum power point tracking under 1 sun illumination of CB-NH₂ based and control PSCs in air (~30% RH and ~40 °C).

4. Line 333: trap densities extracted from SCLC measurements are quoted to three significant figures. Is this level of precision appropriate given sample-sample variation and the assumptions required to extract these values?

We thank the reviewer for pointing this out. Indeed, according to a recent study by Le Corre and co-workers⁹ as well as Siekmann and co-workers¹⁰, the trap density data from SCLC measurement need to be treated carefully when the electrode charge density per cell volume (CU/eV) is higher or equal to the trap density. This is typically the case for thin active layers of 500 nm.¹⁰ We therefore decided to not specify the traps density, rather that the unipolar SCLC characteristics is in principal consistent with the improved V_{oc} : “. As shown and discussed in **Supplementary Fig. S14**, the current in the ohmic region is smaller for the CB-NH₂ based device, which could be correlated with a lower overall trap density and the improved V_{oc} ⁵⁰. However, the measurable trap density from this experiment is limited by the electrode charge per unit cell volume, which is typically very similar to the apparent trap density in thin film (~500 nm) devices as recently shown in ref.^{51,52} ”

5. Line 391: The conductive AFM results show >3x increase in photocurrent for CB-NH₂ samples at 1V bias. It is not obvious whether there is a quantitative relationship between these results and the device performance. In particular, why is the difference in (averaged) nanoscale photocurrent so much larger than the photocurrent observed at the device level?

This is an interesting question. Although there are similarities, there are also significant differences between C-AFM and the JV measurement. First of all, we would like to note that the power of the lamp is only ~5 mW/cm² (not 20 mW/cm² as wrongfully specified in the manuscript, which is now corrected). Therefore, the produced photocurrent is less than the photocurrent under AM1.5G illumination of the solar simulator and the dark current is the dominant factor in the experiment. We have now clarified this in the main text: “In addition, the samples were illuminated with a 5 mW/cm² white LED to generate electron hole pairs.”

In any case, the tip we used for the C-AFM measurement (NSG10 with Pt coating at the tip) has a radius of only 35 nm. This strongly limits the extracted current. In addition, although the gold contact is in contact mode with the C₆₀-terminated film, the configuration of samples measured in this

experiment differs from actual device, which might influence the dynamic of the charge carriers. Therefore, we would not expect the exact same trend as in the *JV*. Nevertheless, the higher signal for electron extraction indicates directional transport of electrons to the top surface in both devices. In addition, the higher electron signal in the case of the CB-NH₂ treated devices means a higher conductivity or lower contact resistance. Regarding the hole extraction, a slightly lower signal is observed in the CB-NH₂ device consistent with the lower dark current. We have now clarified these considerations in the paper: "Note due to the low light intensity (5 mW/cm²) the measurement is in essence a dark *JV* measurement, and the low values of the current are limited by the small radius of the tip of 35 nm. Therefore, the current signal measured from C-AFM is not comparable to *J_{sc}* from *JV* curves."

Finally, we note that this method has been used in Xu¹² and Pietro's paper¹³ which are now cited in the manuscript.

6. There are multiple typos and a few grammatical errors throughout the manuscript. Careful proof-reading is recommended before final submission.

Thank you, we carefully checked the whole manuscript and corrected several spelling mistakes and several sentences, for example:

"The detailed synthesis, a description of the CB-NH₂ molecule and additional molecular structure characterizations (such as nuclear magnetic resonance spectra) can be found in the Supporting Information."

"Any improvement is surprising given the volatile nature of the CB molecule, therefore we speculate there remains a small amount on the surface."

"We use a laser of 520 nm wavelength with a spot size of near 0.5 cm² to illuminate the samples with a 1 sun equivalent intensity by adjusting the produced current close to the *J_{sc}* under a standard solar simulator."

"In addition, electrochemical impedance spectroscopy (EIS) measurements were performed to characterize the charge carrier recombination at open circuit conditions (≈ 1 V)."

Reviewer: 2

Recommendation: insufficient to be published (as noted below).

Comments:

For the "Overcoming C60-induced interfacial recombination in inverted perovskite solar cells by stable electron transporting carborane", the authors introduced carborane as electron transport layer in perovskite solar cell structure. However, the manuscript lacks novelty on the use of carborane and progressive discussion of defect passivation in this study and needs further investigation and improvement of the experimental data. Therefore, this manuscript is insufficient to be published in Nature Communications.

We thank the reviewer for the feedback, however, we disagree with the statement of insufficient novelty, because it is the first time that a carborane based material was applied as electron-selective and passivation layer in perovskites photovoltaics and this particular molecule was never used in any other electronic device. In light of the reviewer comments, we have now improved the discussion of the defect passivation and the interpretation of the experimental data. For example, we further verified the importance of carborane moiety by studying the performance of a phenyl substituted carborane (CB-ph) without the amino group as demonstrated below. We hope that these changes the manuscript will persuade the reviewer to accept the paper.

1. Although carborane is a material that exhibits unique properties, there are more cases of perovskite solar cells applied in previous studies than the authors mentioned. Therefore, authors should add references related to perovskite photovoltaics applied with carborane.

We thank the reviewer for bringing this to our attention. Unfortunately, we actually overlooked one paper where a CB-derivate was used as a part of the hole transporting layer (HTL), however with a different functional group (N,N-bis(4-methoxyphenyl)aniline). The applications of this molecule led to an improved device performance due to an improved FF which was attributed to better carrier transport at the HTL side. We have now cited this paper as ref.¹⁴ However, our work is the first time that a carborane based material is applied as electron/selective passivation layer and this derivate has never been reported before. We focus on the electron transport properties, passivation ability, and elimination of C₆₀-induced interfacial non-radiative recombination, which are unique features of our CB-NH₂ based devices that have not been previously reported. Finally, we also found another interesting paper where the electron-accepting properties of carborane are discussed which is now cited in the main text¹⁵: "They are also tunable as one can functionalize these molecules with various functional groups to tune the electronic structure and other properties of interest^{29,30}. Furthermore, they have previously been used in light-emitting diodes^{31,32} and as building block of hole transporting materials in PSCs to improve the charge transfer rate³³."

2. It is hard to understand that the carborane did not affect the perovskite even though the carborane was deposited on top of the perovskite. Since the carborane has a great ability to absorb neutrons, the carborane can control charge flows and ion movements.

To address the question whether the perovskite can be affected by the CB-NH₂, we first compared the crystallinity and morphology with/without CB-NH₂, we have compared the absorbance of films with and without CB-NH₂ which did not reveal significant changes in the bulk properties as seen from the absorbance spectra (new **Supplementary Fig. S10**). Also, no obvious difference was found from the XRD measurement (**Supplementary Fig. S11**). These results are expected because we use a very low concentration so that the CB-NH₂ layer is very thin (a few nm). However, the PLQY and PL lifetime of perovskite films were remarkably improved upon coating carborane derivatives, as shown in **Figure 4a** and **Supplementary Fig. S14**. Moreover, we detect the presence of B signal in the CB-NH₂ treated perovskite surface by XPS (**Supplementary Fig. S23**) and the variation of binding energy for the Pb signal, which prove the influence of CB-NH₂ on the electronic properties of perovskite surface.

Supplementary Fig. S10. Absorption spectra of control and CB-NH₂ treated films.

Regarding the flow of charges and mobile ions as mentioned by the reviewer. Indeed, our experimental results reveal significant changes in the charge transport, for example the suppression of non-radiative recombination and differences in the extraction of electrons in C₆₀-terminated partial cell stacks with C-AFM. Regarding the impact of mobile ions, although we also agree that the presence of CB-NH₂ at the perovskite surface can also impact the movement of mobile ions and the ion induced losses, we believe that this is outside the scope of the present study which focuses on the impact on non-radiative recombination and stability.

2. As the author mentioned, if only a small amount of carborane remains on top of the perovskite, is the carborane very poorly coated on the perovskite? If coating properties are not secured, it would be a meaningful result to use a material other than CB-NH₂ with an ammonium functional group. Therefore, the authors should present the special properties of carborane alone.

This is a good and important point. It is important to note that the carborane unit is of particular importance in our materials design. Different from many other reported interfacial passivation materials, the carborane derivatives exhibit good electron-transporting properties due to the unique three-dimensional aromaticity. This allows to maintain a high electron transfer across this interface while minimizing interfacial non-radiative recombination through better hole blocking as a result of the deep HOMO level. Unfortunately, as stated in the manuscript, the unmodified carborane undergoes obvious sublimation during annealing and reduced pressure. We speculate that is due to the weak intermolecular forces. To tackle this problem, we attached the phenylamino group to the carborane molecule to increase the intermolecular force and the adsorption to the Pb-terminated (Pb-rich) surface. As such, after incorporation of the phenyl amino group, the film formation and thermal and device stability are greatly enhanced. We note the coating of CB-NH₂ is secured as proven by the presence of Boron on the surface in XPS (**Supplementary Fig. S23**). Moreover, the amine group passivates surface defects. Therefore, the carborane moiety and amino group are working synergistically and are indispensable. Finally, we note that compared to literature, CB-NH₂ outperforms similar passivating molecules. For example, Yang et al. had used a phenethylamine as

passivation layer between perovskite and C60, similar to the functional phenylamine group in CB-NH₂, which however did not result in an improved FF compared to the control device.²⁰ This is likely related to the insulating properties of phenethylamine which restricts the transport of major carriers.

To further emphasize the indispensability of carborane moiety, we compared the performance of CB-ph (without amino) based samples and control samples. The *JV* distribution is shown as below. The CB-ph based samples show a ~10 mV improvement in V_{OC} and a ~2% increased FF compared to the control samples. The improved V_{OC} indicates that the hole blocking effect originates from the carborane moiety, while the improved FF is consistent with the improved electron extraction seen in C-AFM.

Supplementary Fig. S21. Parameters distribution of V_{OC} , J_{SC} , FF, and PCE for control, and CB-ph treated devices, respectively.

We also conducted additional PLQY measurements on partial cell stacks to evaluate the effect of CB-ph on the non-radiative recombination process. As shown in **Supplementary Fig. S22** below, for the ITO/HTL/pero/C60 stack, we notice a ~10 mV improvement in the quasi-Fermi level splitting, but for the ITO/HTL/pero stack, we did not see any difference with and without CB-ph. This confirms the ability of carborane to reduce non-radiative recombination between perovskite and C60 without the amino group. However, the unchanged QFLS for ITO/HTL/pero stack indicates that there is no passivation effect from carborane. This is in contrast to the CB-NH₂ molecules which also passivate the perovskite surface.

Supplementary Fig. S22. Photoluminescence spectrum and the calculated photoluminescence quantum yield measurements (PLQY) and internal quasi-Fermi level splitting (QFLS) with/without CB-ph.

Overall, the results suggest that there is some contribution of the improved hole blocking to the total V_{oc} gain with the CB-NH₂ functionalized device (50 mV), although the effect of the amino group is more significant. As for the FF improvement, our results indicate a similar improvement between CB-ph and CB-NH₂, which suggests that the improvement originates from the hole blocking and electron extraction ability of the CB molecule. We have now included this discussion in a new section in the revised manuscript:

Contribution to device performance from carborane and amino moiety. In order to investigate the contribution from the carborane and amino group moiety in CB-NH₂ to the improvement of device performance, we also studied the intermediate phenyl functionalized carborane (CB-ph) without the amino group because the bare carborane suffers from sublimation on the perovskite layer. As can be seen in **Supplementary Fig. S21**, CB-ph leads to a significant FF improvement (~2%) but only slight V_{oc} improvement (~10 mV), indicating that the carborane alone reduces non-radiative recombination less effectively than CB-NH₂ but has a significant effect on the electron extraction which is consistent with the C-AFM result. This conclusion can be further confirmed by the improved PLQY of the ITO/perovskite/C₆₀ stack with the CB-ph interlayer but identical PLQY of the ITO/perovskite with and without the CB-ph interlayer (**Supplementary Fig. S22**). These results suggest that there is a small contribution (~10 mV) of the carborane moiety to the total V_{oc} gain of the CB-NH₂ functionalized device (~50 mV) which we attribute to the improved hole blocking ability of the carborane. However, the effect of the amino group is more significant. As for the FF improvement, our results indicate a more significant contribution of the carborane moiety (~2%) to the total gain (~4%), which suggests that the improvement originates partially from the hole blocking and electron extraction ability of the CB molecule. Nevertheless, considering that the packing and adsorption ability of the CB-ph and the CB-NH₂ molecules are likely different, the exact contribution of the carborane

moiety is difficult to quantify. Given that the performance of the CB-NH₂ device is optimum, we can conclude that the carborane and the amino group are working synergistically and are indispensable.”

3. Moreover, it has been reported in previous studies that inhibition of non-radiative recombination has a great effect on J_{sc} as well as V_{oc} . In this study, CB-NH₂ reduced the loss occurring at the interface through defect passivation. However, defect states will also impede the flow of photogenerated charges. Why is the effect at J_{sc} smaller than the effect at V_{oc} ?

Here we respectfully disagree with the reviewer. Although, non-radiative recombination can in principal effect the J_{sc} , as long as the JV curve is flat until the maximum power point, an impact of non-radiative recombination is not expected. Only at the point where the current starts to decrease (at forward bias voltages close to the maximum power point). The simulation based on the model described in the work by Diekmann et al.²¹ shows the impact of a 10-fold reduced density at the perovskite/C60 interface compared to the control, **Figure R1**. In panel **b**, the corresponding recombination current are shown for both cases, passivated and control. It can be seen that the recombination currents, in particular the interfacial recombination, are strongly voltage dependent and increase with applied forward bias. We note, that bulk recombination can be more independent with voltage in case of nearly flat energy bands under short-circuit conditions. In this case, non-radiative recombination can be significant at 0V, however, an optimization of the interface, as it was done in this work, will not affect these bulk losses.

Figure R1. **a** Simulated current-voltage characteristics of cells with and without a passivation at the perovskite/C60 interface which reduces the interfacial defect density by a factor of 10. **b** Corresponding parallel recombination currents for the 2 simulated cells shown in panel a.

Reviewer: 3

Recommendation: Publish after addressing the following concerns

Comments:

The authors report a novel interlayer material, which helps to achieve efficient and stable inverted perovskite solar cells. In detail, o-carborane decorated with phenylamino groups (CB-NH₂) was introduced on the perovskite surface to eliminate the non-radiative recombination loss across the perovskite/C60 interface. The results are interesting although some of the claims along with the work novelty are overrated. This manuscript shows almost all the data I

want to see. Therefore, I suggest accepting the manuscript after addressing the following concerns.

We thank the reviewer for the critical feedback and the recommendation to publish our work.

1. The authors introduced a novel material CB-NH₂, so is it the unique advantage of CB or the amino group that plays the leading role? The authors claim that CB-NH₂ can not only passivate surface defects (due to bonding effect from N-H...I and N-Pb), but also block holes (due to the type-I band alignment). That is to say, CB-NH₂ can reduce both the non-radiative recombination assisted by surface defects and the carrier quenching at the perovskite/C60 interface. Which effect contributes more to reducing VOC losses? Since the synthesized intermediates contain CB-ph and CB-NO₂ (almost no bonding effect with perovskite), I wonder how they affect the device performance.

We thank the reviewer for raising this important point. The reviewer is correct that we interpret the improved performance with the CB-NH₂ functionalization based on the synergistic working mechanism of the carborane moiety and the amino group, where the former passivates defects, while the latter improves the hole blocking due to the deep HOMO energy level, while providing sufficient electron transport (in contrast to many other commonly used molecules).

We note that this important comment also goes back to *comment 2 of Reviewer 2* which has been addressed above. Briefly again, in order to understand the relative impact of the carborane moiety without the amino group as suggested by the reviewer, we fabricated a phenyl functionalized carborane molecules (CB-ph without the amino group) and carefully compared the *JV* performance metric and PLQY with the control devices. We studied the CB-ph instead of the bare carborane as the latter suffers from the problem of sublimation, thus the film quality is not secured. As shown above in response to *Reviewer 2*, the *JV* and PLQY results (**Supplementary Fig. S21** and **S22**) suggest that there is some contribution (~10%) of the improved hole blocking due carborane moiety to the total V_{oc} gain with the CB-NH₂ functionalized device (~50 mV), although the effect of the amino group is more significant. As for the FF improvement, our results indicate a similar improvement between CB-ph and CB-NH₂, which suggests that the improvement originates from the hole blocking and electron extraction ability of the CB molecule. We have also included these considerations in in a new section in the revised manuscript.

Lastly, regarding the CB-NO₂ molecule mentioned by the reviewer, we actually also tried this molecule, however, the performance was strongly negatively affected (**Figure R2**). We believe that this is due to the NO₂ which has been linked to increasing recombination in previous works.

Figure R2. *JV* characteristics and performance metric of devices treated with nitrobenzene and phenyl substituted carboranes (CB-NO₂ and CB-ph, respectively).

2. As type-I band alignment (as shown in Fig. 1c), why does perovskite/CB-NH₂ increase electron extraction?

We thank the reviewer for raising this interesting question. Although the C-AFM does show an improved electron extraction the underlying mechanism is not clear. First, we note that the energy level depicted in **Figure 1c** represent the energy levels measured on individual layers, therefore we expect the energy alignment in the device to be different. Apart from the exact energy alignment at the interface, we believe a plausible explanation for the improved electron extraction observed in the C-AFM measurement might stem from reduced defect density at the interface as trapped electrons can repel the extraction of following electrons. Evidence for the reduced trap states is provided in the manuscript, for example from the PLQY and *JV* results, while the FF of the CB-NH₂ treated device is also improved which is consistent with the improved electron extraction. We now, included this possible interpretation in the manuscript.

In addition, we also measured ultraviolet photoelectron spectroscopy (UPS) of the perovskite/C60 (thickness from 1nm to 30 nm) film with/without CB-NH₂. As shown in **Figure R3** below, with increasing C60 thickness up to 30 nm, we observed a significant upwards bend banding in the C60 layer in the control sample, which can be explained and modelled by a *n*-doped perovskite surface. In contrast, in the perovskite/CB-NH₂/C60 films, we did not detect obvious energy level offset on thin and thick C60 samples. Moreover, CB-NH₂ raises the conduction band of the perovskite which better aligns with the perovskite bulk energy level. We note that the perovskite energy levels only represent the energetics at the surface of the perovskite accessible to UPS, thus not representative of the bulk. Although, the interpretation of these results is still ongoing and thus not presented in the main text, these results may provide further clues as to why CB-NH₂ improves the electron transporting.

Figure R3. Energy alignment at the perovskite C60 interface as deduced from ultraviolet photoelectron spectroscopy (UPS) of films with a variable C60 thickness. The graph demonstrates a significant flattening of the C60 bend bending in case of CB-NH₂ which could be linked to the better charge extraction although further investigations are required towards a quantitative description of the interfacial energy levels. We note, the perovskite energy levels only represent the energetics at the surface of the perovskite accessible to UPS, thus not representative of the bulk.

3. “As shown in Fig. 2e, the stabilized current density of control and CB-NH₂ treated cell are 23.73, 23.26 mA/cm² under a bias of 0.97 and 0.91V, respectively, confirming a stabilized PCE of 23.04% and 21.10%, which are consistent with *JV* results.” The data for control and CB-NH₂ treated cell were reversed.

Thank you very much, indeed the data for control and CB-NH₂ treated cell were reversed. we revised the sentence as follow: “As shown in **Figure 2e**, the stabilized current density of CB-NH₂ treated and control devices are 23.73, 23.26 mA/cm² under a bias of 0.97 and 0.91V, respectively, confirming a stabilized PCE of 23.04% and 21.10%, which are consistent with *JV* results.”

4. “To quantify the stability, we performed MPP tracking on the devices in a glovebox... The improved operational stability is attributed to... as well as the increased hydrophobicity and resistance to moisture.” Stability results obtained in the glovebox should not be related to moisture.

We agree with the reviewer that this measurement does not allow us to conclude about the benefit of the increase hydrophobicity on the stability of the devices and we have now removed the conflicting sentence from the manuscript. In light of *comment 3* from *Reviewer 1*, we also performed another maximum power point tracking measurement of devices outside the glovebox in air with a relative humidity of ~30% RH and a temperature of ~40 °C. As shown in **Supplementary Fig. S12**, the encapsulated CB-NH₂ is more stable than the control device which strongly points to the benefit of a more hydrophilic interface for the device stability.

5. The caption of Fig. S15 is incorrect. In addition, the resolution of light intensity up to 0.0001 sun is incredible. Please provide the corresponding J-V curves of the devices under different light intensity in Fig S15.

We thank the reviewer for pointing out this mistake which we have now corrected. We note that **Supplementary Fig. S15 (Supplementary Fig. S19** in revised Supplementary Information) only shows the V_{OC} as a function of light intensity, which we measure by continuously decreasing the intensity using an automated filter wheel as shown below (**Figure R4**). For each intensity we measure the V_{OC} for ~ 2 seconds before measuring the J_{SC} for the same time (J_{SC} not plotted in this graph). Then the filter wheel is moved to the next position and the routine is repeated. We have now provided experimental details in the Methods section. We also note that although, these measurements were conducted by measuring V_{OC} and J_{SC} only, we can also measure the full JV curves over the same intensity range with the same setup.

Figure R4. Filter wheel and laser used for automatic V_{OC} measurements.

References:

1. Warby, J. *et al.* Understanding Performance Limiting Interfacial Recombination in *pin* Perovskite Solar Cells. *Advanced Energy Materials* 2103567 (2022) doi:10.1002/aenm.202103567.
2. Wang, Q., Dong, Q., Li, T., Gruverman, A. & Huang, J. Thin Insulating Tunneling Contacts for Efficient and Water-Resistant Perovskite Solar Cells. *Advanced Materials* **28**, 6734–6739 (2016).
3. Peng, J. *et al.* A Universal Double-Side Passivation for High Open-Circuit Voltage in Perovskite Solar Cells: Role of Carbonyl Groups in Poly(methyl methacrylate). *Advanced Energy Materials* **8**, 1801208 (2018).

4. Zhuang, Q. *et al.* Enhanced Performance and Stability of TiO₂-Nanoparticles-Based Perovskite Solar Cells Employing a Cheap Polymeric Surface Modifier. *ChemSusChem* **12**, 4824–4831 (2019).
5. Wolff, C. M. *et al.* Reduced Interface-Mediated Recombination for High Open-Circuit Voltages in CH₃NH₃PbI₃ Solar Cells. *Adv. Mater.* **29**, 1700159 (2017).
6. Rakstys, K. *et al.* Triazatruxene-Based Hole Transporting Materials for Highly Efficient Perovskite Solar Cells. *J. Am. Chem. Soc.* **137**, 16172–16178 (2015).
7. Xu, B. *et al.* Carbazole-Based Hole-Transport Materials for Efficient Solid-State Dye-Sensitized Solar Cells and Perovskite Solar Cells. *Advanced Materials* **26**, 6629–6634 (2014).
8. Wang, Y. *et al.* Optical Gaps of Organic Solar Cells as a Reference for Comparing Voltage Losses. *Advanced Energy Materials* **8**, 1801352 (2018).
9. Le Corre, V. M. *et al.* Revealing Charge Carrier Mobility and Defect Densities in Metal Halide Perovskites via Space-Charge-Limited Current Measurements. *ACS Energy Lett.* **6**, 1087–1094 (2021).
10. Siekmann, J., Ravishankar, S. & Kirchartz, T. Apparent Defect Densities in Halide Perovskite Thin Films and Single Crystals. *ACS Energy Lett.* **6**, 3244–3251 (2021).
11. Tan, H. *et al.* Efficient and stable solution-processed planar perovskite solar cells via contact passivation. *Science* **355**, 722–726 (2017).
12. Xu, J. *et al.* Perovskite–fullerene hybrid materials suppress hysteresis in planar diodes. *Nat Commun* **6**, 7081 (2015).
13. Caprioglio, P. *et al.* Bi-functional interfaces by poly(ionic liquid) treatment in efficient pin and nip perovskite solar cells. *Energy Environ. Sci.* **14**, 4508–4522 (2021).
14. Kim, B. G. *et al.* An unusual charge transfer accelerator of monomolecular Cb-OMe (4,4'-(ortho-carborane)bis(N,N-bis(4-methoxyphenyl)aniline) in perovskite optoelectronic devices. *Solar Energy Materials and Solar Cells* **208**, 110414 (2020).
15. Lee, S., Shin, J., Ko, D.-H. & Han, W.-S. A new type of carborane-based electron-accepting material. *Chem. Commun.* **56**, 12741–12744 (2020).

16. Dash, B. P., Satapathy, R., Gaillard, E. R., Maguire, J. A. & Hosmane, N. S. Synthesis and Properties of Carborane-Appended C_3 -Symmetrical Extended π Systems. *J. Am. Chem. Soc.* **132**, 6578–6587 (2010).
17. King, R. B. Three-dimensional aromaticity in deltahedral boranes and carboranes. *Russ Chem Bull* **42**, 1283–1291 (1993).
18. Wee, K.-R. *et al.* Carborane-Based Optoelectronically Active Organic Molecules: Wide Band Gap Host Materials for Blue Phosphorescence. *J. Am. Chem. Soc.* **134**, 17982–17990 (2012).
19. Furue, R., Nishimoto, T., Park, I. S., Lee, J. & Yasuda, T. Aggregation-Induced Delayed Fluorescence Based on Donor/Acceptor-Tethered Janus Carborane Triads: Unique Photophysical Properties of Nondoped OLEDs. *Angewandte Chemie International Edition* **55**, 7171–7175 (2016).
20. Yang, S. *et al.* Tailoring Passivation Molecular Structures for Extremely Small Open-Circuit Voltage Loss in Perovskite Solar Cells. *J. Am. Chem. Soc.* **141**, 5781–5787 (2019).
21. Diekmann, J. *et al.* Pathways toward 30% Efficient Single-Junction Perovskite Solar Cells and the Role of Mobile Ions. *Solar RRL* **5**, 2100219 (2021).

REVIEWER COMMENTS

Reviewer #1 (Remarks to the Author):

I appreciate the significant effort by the authors to address my comments and those of the other reviewers. The responses and manuscript amendments adequately address all of the points raised in my previous review, so I am happy to recommend the manuscript for publication without further changes.

Reviewer #2 (Remarks to the Author):

The authors prepared to response the reviewers' questions by faithfully performing additional experiments requested by the reviewers. The manuscript has improved in the first review round. However, the material effect of carborane is still not clear and the novelty seems pretty lack. Many previous studies have reported an increase in Voc due to penetration of PCBM and C60 along the grain boundaries of the perovskite.

It is thought that the carborane in this study will also have the effect of increasing the interface by the carborane infiltrating along the grain boundary as in the previous study. Since the mobility of electrons is affected by the difference in electron affinity between C60 and carborane, it is recommended to check electron deflection through DFT simulation.

Also, reviewers doubt whether it is difficult to drive the carborane on its own without the C60. It is expected that the effects of carborane, which the authors mention, can replace as well as supplement the C60. The reviewer is wondering if the perovskite device with only carborane as thick as C60, which has secured coating, works.

Reviewer #3 (Remarks to the Author):

The authors have addressed my previous concern, and therefore I recommend its publication in Nature Communications.

Point-by-point Response to Reviewer(s)' Comments:

Reviewer: 1

I appreciate the significant effort by the authors to address my comments and those of the other reviewers. The responses and manuscript amendments adequately address all of the points raised in my previous review, so I am happy to recommend the manuscript for publication without further changes.

We are delighted to receive such a positive feedback, thank you very much.

Reviewer: 2

The authors prepared to response the reviewers' questions by faithfully performing additional experiments requested by the reviewers. The manuscript has improved in the first review round.

We appreciate that the reviewer acknowledges our efforts with the last revision.

However, the material effect of carborane is still not clear and the novelty seems pretty lack. Many previous studies have reported an increase in Voc due to penetration of PCBM and C60 along the grain boundaries of the perovskite. It is thought that the carborane in this study will also have the effect of increasing the interface by the carborane infiltrating along the grain boundary as in the previous study

We would like to note that the observed improvement of the device performance is not related to the penetration of the grain boundaries with PCBM, or C₆₀ nor increasing the interface [area?] by the infiltration of carborane along the grain boundaries. In fact, we would disagree with both of these implied conclusions of the reviewer. Compared to the neat perovskite with its grain boundaries, we have consistently shown that the deposition of C₆₀ and PCBM actually lowers the open-circuit voltage potential or quasi-Fermi level splitting¹⁻³. This is due to interfacial recombination losses which greatly outweigh the recombination at the grain boundaries and at the perovskite surface. A large fraction of the community now agrees on this general conclusion⁴⁻⁶. We also disagree with the postulation that carborane increases the interfacial area thereby reducing recombination, not least because the interface area needs to be reduced to have a positive effect. As highlighted before, the positive synergic effect of the here functionalized carborane CB-NH₂ as an exemplary molecule of a novel class of electron transport layers for various optoelectronic perovskite devices is to block the holes from the C₆₀-interface via its deep HOMO level and by passivating traps on the neat perovskite while maintaining a high electron extraction rate. To proof these points we have provided substantial additional data during the last revision which leads to different conclusions than suggested by the reviewer. Therefore, in our opinion, these reviewer's comments do not challenge the novelty of our paper.

Since the mobility of electrons is affected by the difference in electron affinity between C₆₀ and carborane, it is recommended to check electron deflection through DFT simulation.

We would like to point out that electron mobility is independent of the energy levels and is thus not related to different electron affinity between C₆₀ and carborane. The relatively high LUMO of the carborane may reduce the (effective) electron conductivity in the C₆₀ layer in the device by allowing less charges through. However, as we have demonstrated with selective conductive AFM-measurements, this is not the case experimentally for a thin layer of CB-NH₂ as the extraction of electrons through the C₆₀ is actually improved with respect to C₆₀-only as shown by conductive AFM. As noted before, this could be related to the reduced defect density at the surface as trapped electrons could repel the extraction of the remaining ones. Moreover, our UPS measurements shown in the last rebuttal letter indicate a better energy alignment between the perovskite and C₆₀ with the carborane. Finally, regarding the suggestion to check the “electron deflection” with DFT simulation, we have discussed this internally but unfortunately we do not know what was meant with electron deflection. We believe that our DFT simulations already provide substantial additional evidence of the strong adsorption of the CB-NH₂ layer on the perovskite surface and the passivation effect of the amino moiety. Therefore, we do not think that more DFT simulations will provide further insights at this point.

Also, reviewers doubt whether it is difficult to drive the carborane on its own without the C₆₀. It is expected that the effects of carborane, which the authors mention, can replace as well as supplement the C₆₀. The reviewer is wondering if the perovskite device with only carborane as thick as C₆₀, which has secured coating, works.

Although we agree with the reviewer that it would be indeed nice to be able to entirely replace the C₆₀ with a carborane, we have not claimed to do so with this particular carborane in this paper, rather we employ it as a thin interlayer as stated in the manuscript. To address the reviewer comment, we prepared a batch of devices to replace C₆₀ with a thicker layer of CB-NH₂, which however showed poorer performance. We believe that this is due to the still lower mobility of the carboranes compared to C₆₀. Nevertheless, it is important to point out that carboranes are highly tuneable allowing to attach various functionalized groups, which will undoubtedly allow to tailor the properties of this exciting class of molecules. Notwithstanding this point, this manuscript clearly demonstrates various other beneficial properties of the CB-NH₂ molecules, which already allows to essentially overcome the C₆₀-induced interfacial recombination, which has been a major limiting factor of *pin*-type cells.

Reviewer: 3

The authors have addressed my previous concern, and therefore I recommend its publication in Nature Communications.

We thank the reviewer for the recommendation to publish our work.

References

1. Warby, J. *et al.* Understanding Performance Limiting Interfacial Recombination in pin Perovskite Solar Cells. *Advanced Energy Materials* **12**, 2103567 (2022).
2. Stolterfoht, M. *et al.* The impact of energy alignment and interfacial recombination on the internal and external open-circuit voltage of perovskite solar cells. *Energy Environ. Sci.* **12**, 2778–2788 (2019).
3. Stolterfoht, M. *et al.* Visualization and suppression of interfacial recombination for high-efficiency large-area pin perovskite solar cells. *Nat Energy* **3**, 847–854 (2018).
4. Cacovich, S. *et al.* Imaging and quantifying non-radiative losses at 23% efficient inverted perovskite solar cells interfaces. *Nat Commun* **13**, 2868 (2022).
5. Li, L. *et al.* Flexible all-perovskite tandem solar cells approaching 25% efficiency with molecule-bridged hole-selective contact. *Nat Energy* **7**, 708–717 (2022).
6. Liu, J. *et al.* Efficient and stable perovskite-silicon tandem solar cells through contact displacement by MgFx. *Science* **377**, 302–306 (2022).